# Transcription factor clusters regulate genes in eukaryotic cells

**Adam JM Wollman[1][†], Sviatlana Shashkova[1,2][†], Erik G Hedlund[1], Rosmarie Friemann[2], Stefan Hohmann[2,3], Mark C Leake[1]\***

[1]Biological Physical Sciences Institute, University of York, York, United Kingdom; [2]Department of Chemistry and Molecular Biology, University of Gothenburg, Gothenburg, Sweden; [3]Department of Biology and Biological Engineering, Chalmers University of Technology, Göteborg, Sweden

**Abstract** Transcription is regulated through binding factors to gene promoters to activate or repress expression, however, the mechanisms by which factors find targets remain unclear. Using single-molecule fluorescence microscopy, we determined in vivo stoichiometry and spatiotemporal dynamics of a GFP tagged repressor, Mig1, from a paradigm signaling pathway of *Saccharomyces cerevisiae*. We find the repressor operates in clusters, which upon extracellular signal detection, translocate from the cytoplasm, bind to nuclear targets and turnover. Simulations of Mig1 configuration within a 3D yeast genome model combined with a promoter-specific, fluorescent translation reporter confirmed clusters are the functional unit of gene regulation. In vitro and structural analysis on reconstituted Mig1 suggests that clusters are stabilized by depletion forces between intrinsically disordered sequences. We observed similar clusters of a co-regulatory activator from a different pathway, supporting a generalized cluster model for transcription factors that reduces promoter search times through intersegment transfer while stabilizing gene expression.

DOI: https://doi.org/10.7554/eLife.27451.001

**\*For correspondence:**
mark.leake@york.ac.uk

[†]These authors contributed equally to this work

**Competing interests:** The authors declare that no competing interests exist.

## Introduction

Cells respond to their environment through gene regulation involving protein transcription factors. These proteins bind to DNA targets of a few tens of base pairs (bp) length inside ~500–1,000 bp promoter sequences to repress/activate expression, involving single (*Jacob and Monod, 1961*) and multiple (*Gertz et al., 2009*) factors, resulting in the regulation of target genes. The mechanism for finding targets in a genome ~six orders of magnitude larger is unclear since free diffusion followed by capture is too slow to account for observed search times (*Berg et al., 1981*). Target finding may involve heterogeneous mobility including nucleoplasmic diffusion, sliding and hops along DNA up to ~150 bp, and even longer jumps separated by hundreds of bp called intersegment transfer (*Mahmutovic et al., 2015*; *Halford and Marko, 2004*; *Gowers and Halford, 2003*).

In eukaryotes, factor localization is dynamic between nucleus and cytoplasm (*Whiteside and Goodbourn, 1993*). Although target binding sites in some cases are known to cluster in hotspots (*Harbison et al., 2004*) the assumption has been that factors themselves do not function in clusters but as single molecules. Realistic simulations of diffusion and binding in the complex milieu of nuclei suggest a role for multivalent factors to facilitate intersegment transfer by enabling DNA segments to be connected by a single factor (*Schmidt et al., 2014*).

The use of single-molecule fluorescence microscopy to monitor factor localization in live cells has resulted in functional insight into gene regulation (*Li and Xie, 2011*). Fluorescent protein reporters, in particular, have revealed complexities in mobility and kinetics in bacterial (*Hammar et al., 2012*)

and mammalian cells (*Gebhardt et al., 2013*) suggesting a revised view of target finding (*Mahmutovic et al., 2015*).

Key features of gene regulation in eukaryotes are exemplified by glucose sensing in budding yeast, *Saccharomyces cerevisiae*. Here, regulation is achieved by factors which include the Mig1 repressor, a Zn finger DNA binding protein (*Nehlin et al., 1991*) that acts on targets including *GAL* genes (*Frolova et al., 1999*). Mig1 is known to localize to the nucleus in response to increasing extracellular glucose (*De Vit et al., 1997*), correlated to its dephosphorylation (*Bendrioua et al., 2014*; *Shashkova et al., 2017*). Glucose sensing is particularly valuable for probing gene regulation since the activation status of factors such as Mig1 can be controlled reproducibly by varying extracellular glucose. Genetic manipulation of the regulatory machinery is also tractable, enabling native gene labeling with fluorescent reporters for functioning imaging studies.

We sought to explore functional spatiotemporal dynamics and kinetics of gene regulation in live *S. cerevisiae* cells using its glucose sensing pathway as a model for signal transduction. We used single-molecule fluorescence microscopy to track functional transcription factors with millisecond sampling to match the mobility of individual molecules. We were able to quantify composition and dynamics of Mig1 under physiological and perturbed conditions which affected its possible phosphorylation state. Similarly, we performed experiments on a protein called Msn2, which functions as an activator for some of Mig1 target genes (*Lin et al., 2015*) but controlled by a different pathway. By modifying the microscope we were also able to determine turnover kinetics of transcription factors at their nuclear targets.

The results, coupled to models we developed using chromosome structure analysis, indicated unexpectedly that the functional component which binds to promoter targets operates as a cluster of transcription factor molecules with stoichiometries of ~6–9 molecules. We speculated that these functional clusters in live cells were stabilized through interactions of intrinsically disordered sequences facilitated through cellular depletion forces. We were able to mimic those depletion forces in in vitro single-molecule and circular dichroism experiments using a molecular crowding agent. Our novel discovery of factor clustering has a clear functional role in facilitating factors finding their binding sites through intersegment transfer, as borne out by simulations of multivalent factors (*Schmidt et al., 2014*); this addresses a long-standing question of how transcription factors efficiently find their targets. This clustering also functions to reduce off rates from targets compared to simpler monomer binding. This effect improves robustness against false positive detection of extracellular chemical signals, similar to observations for the monomeric but multivalent bacterial LacI repressor (*Mahmutovic et al., 2015*). Our findings potentially reveal an alternative eukaryotic cell strategy for gene regulation but using an entirely different structural mechanism.

## Results

### Single-molecule imaging reveals in vivo clusters of functional Mig1

To explore the mechanisms of transcription factor targeting we used millisecond Slimfield single-molecule fluorescence imaging (*Plank et al., 2009*; *Reyes-Lamothe et al., 2010*; *Badrinarayanan et al., 2012*; *Miller et al., 2017*) on live *S. cerevisiae* cells (*Figure 1A* and *Figure 1—figure supplement 1*). We prepared a genomically encoded green fluorescent protein (GFP) reporter for Mig1 (*Table 1*). To enable nucleus and cell body identification we employed mCherry on the RNA binding nuclear protein Nrd1. We measured cell doubling times and expression to be the same within experimental error as the parental strain containing no fluorescent protein (*Figure 1—figure supplement 2A*). We optimized Slimfield for single-molecule detection sensitivity with an in vitro imaging assay of surface-immobilized purified GFP (*Leake et al., 2006*) indicating a brightness for single GFP molecules of ~5000 counts on our camera detector (*Figure 1—figure supplement 2B*). To determine any fluorescent protein maturation effects we performed cell photobleaching while expression of any additional fluorescent protein was suppressed by antibiotics, and measured subsequent recovery of cellular fluorescence <15% for fluorescent protein components, corrected for any native autofluorescence, over the timescale of imaging experiments (*Figure 1—figure supplement 2C and D*).

Under depleted (0%)/elevated (4%) extracellular glucose (-/+), we measured cytoplasmic and nuclear Mig1 localization bias respectively, as reported previously (*De Vit et al., 1997*), visible in

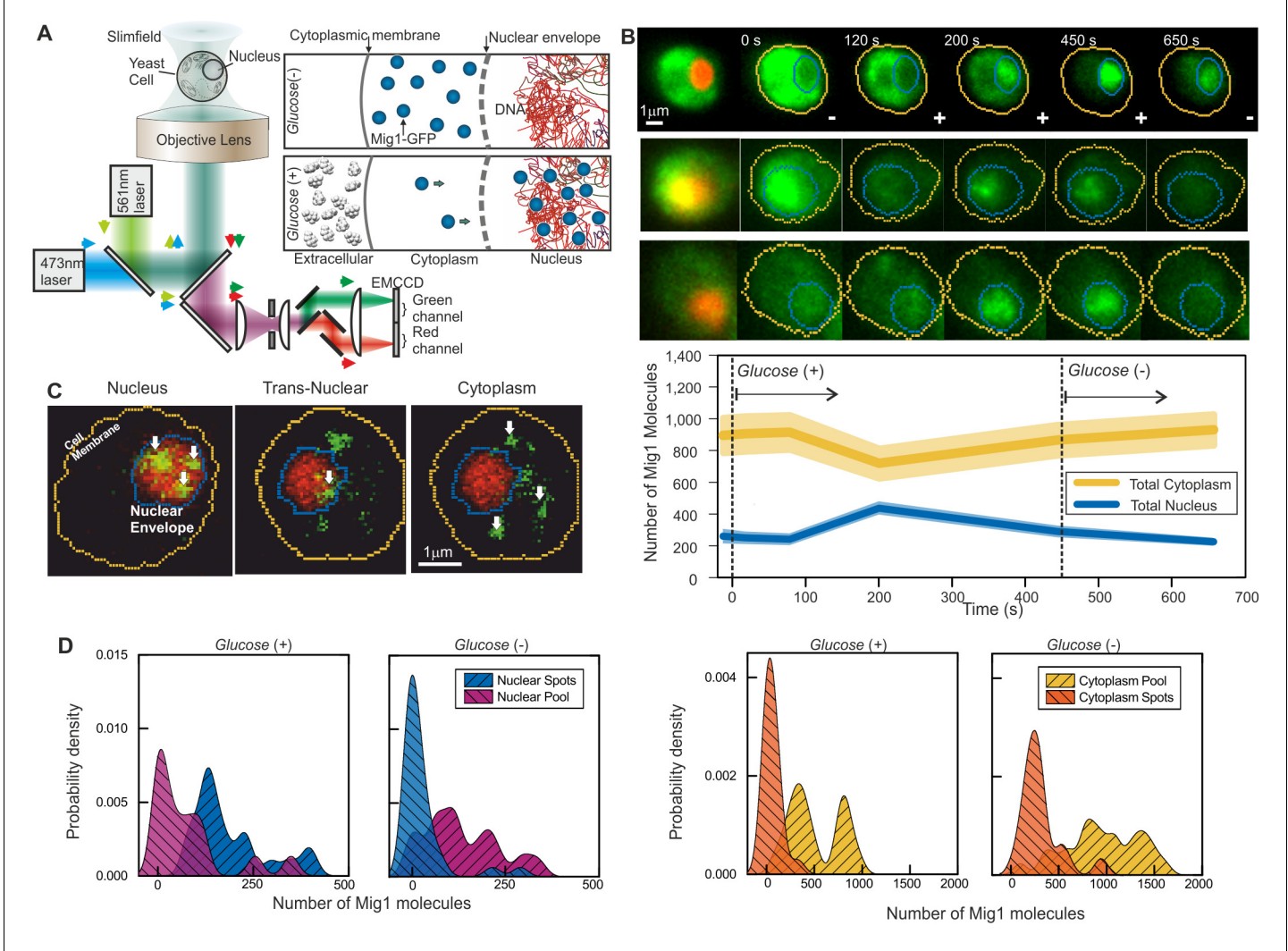

**Figure 1.** Single-molecule Slimfield microscopy of live cells reveals Mig1 clusters. (**A**) Dual-color fluorescence microscopy assay. Mig1-GFP localization change (cyan, right panels) depending on glucose availability. (**B**) Example Slimfield micrographs of change of Mig1-GFP localization (green) with glucose for three cells, nuclear Nrd1-mCherry indicated (red, left), mean and SEM errorbounds of total cytoplasmic (yellow) and nuclear (blue) contributions shown (lower panel), n = 15 cells. Display scale fixed throughout each time course to show pool and foci fluorescence. (**C**) Example Slimfield micrographs of cells showing nuclear (left), trans-nuclear (center) and cytoplasmic (right) Mig1-GFP localization (green, distinct foci white arrows), Nrd1-mCherry (red) and segmented cell body (yellow) and nuclear envelope (blue) indicated. Display scales adjusted to only show foci. (**D**) Kernel density estimations (KDE) for Mig1-GFP content in pool and foci for cytoplasm and nucleus at *glucose* (+/-), n = 30 cells.

DOI: https://doi.org/10.7554/eLife.27451.002

The following figure supplements are available for figure 1:

**Figure supplement 1.** Brightfield and fluorescence micrographs of key strains and glucose conditions.

DOI: https://doi.org/10.7554/eLife.27451.003

**Figure supplement 2.** Fluorescent reporter strains have similar viability to wild type, with relatively fast maturation of fluorescent protein, and no evidence for GFP-mediated oligomerization.

DOI: https://doi.org/10.7554/eLife.27451.004

**Figure supplement 3.** In vivo Mig1-GFP foci intensity traces as a function of time.

DOI: https://doi.org/10.7554/eLife.27451.005

individual cells by rapid microfluidic exchange of extracellular fluid (*Figure 1B*), with high cell-cell variability (*Figure 1B* middle panel). However, our ultrasensitive imaging resolved two novel components under both conditions consistent with a diffuse monomer pool and distinct multimeric foci which could be tracked up to several hundred milliseconds (*Figure 1C* and *Figure 1—figure*

**Table 1.** *S. cerevisiae* cell strains and plasmids.
List of all strains and plasmids used in this study.

| Strain name | Background | Genotype | Source/Reference |
|---|---|---|---|
| YSH1351 | S288C | MATa HIS3D0 LEU2D1 MET15D0 URA3D0 | S. Hohmann collection |
| YSH1703 | W303-1A | MATa mig1Δ::LEU2 snf1Δ::KanMX | S. Hohmann collection |
| YSH2267 | BY4741 | MATa his3D1 leu2D0 met15d0 ura3D0 mig1Δ::KanMX NRD1-mCherry-hphNT1 | S. Hohmann collection |
| YSH2350 | BY4741 | MATa MSN2-GFP-HIS3 NRD1-mCherry-hphNT1 MET LYS | (*Babazadeh et al., 2013*) |
| YSH2856 | BY4741 | MATa MIG1-eGFP-KanMX NRD1-mCherry-HphNT1 snf1Δ::LEU2 MET LYS | *This study* |
| YSH2348 | BY4741 | MATa MIG1-GFP-HIS3 NRD1-mCherry-hphNT1 MET LYS | (*Bendrioua et al., 2014*) |
| YSH2862 | BY4741 | MATa MIG1-GFPmut3-HIS3 | *This study* |
| YSH2863 | BY4741 | MATa MIG1-GFPmut3-HIS3 NRD1-mCherry-HphMX4 | *This study* |
| YSH2896 | BY4741 | MATa MIG1-mEOs2-HIS3 | *This study* |
| ME404 | BY4741 | 'BY4741 MSN2-mKO2::LEU2 MIG1- mCherry::spHIS5 GSY1-24xPP7::KANMX msn4Δ mig2Δ nrg1::HPHMX nrg2::Met15 SUC2::NatMX' | (*Lin et al., 2015*) |
| ME412 | BY4741 | BY4741 MSN2-mKO2::LEU2 MIG1(Δaa36-91)- mCherry::spHIS5 GSY1-24xPP7::KANMX msn4Δ mig2Δnrg1:: HPHMX nrg2::Met15 | (*Lin et al., 2015*) |
| ME411 | BY4741 | MIG1(Δaa36-91)-mCherry::spHIS5 GSY1-24xPP7::KANMX msn4Δ mig2Δnrg1::HPHMX nrg2::Met15 | (*Lin et al., 2015*) |

| Plasmid name | Description | Source/Reference |
|---|---|---|
| pMIG1-HA | HIS3 | (*Schmidt and McCartney, 2000*) |
| pSNF1-TAP | URA3, in pRS316 | S. Hohmann collection |
| pSNF1-I132G-TAP | URA3, in pRS316 | S. Hohmann collection |
| pmGFPS | HIS3, GFPmut3 S65G, S72A, A206K | *This study* |
| pMig1-mGFP | 6xHIS-Mig1-GFPmut3 in pRSET A | *This study* |
| pmEOs2 | mEOs2-HIS3 in pMK-RQ | *This study* |
| YDp-L | LEU2 | (*Berben et al., 1991*) |
| YDp-H | HIS3 | (*Berben et al., 1991*) |
| BM3726 | Mig1 (Ser222,278,311,381 → Ala), URA3, in pRS316 | M. Johnston collection (*DeVit and Johnston, 1999*) |
| pDZ276 | PP7-2xGFP::URA3 | (*Lin et al., 2015*) |

DOI: https://doi.org/10.7554/eLife.27451.006

*Supplement 3*; *Videos 1* and *2*). We wondered if the presence of foci was an artifact due to GFP oligomerization. To discourage artifactual aggregation we performed a control using another type of GFP containing an A206K mutation (denoted GFPmut3 or mGFP) known to inhibit oligomerization (*Zacharias et al., 2002*). However, both in vitro experiments using purified GFP and mGFP (*Figure 1—figure supplement 2B*) and live cell experiments at *glucose* (-/+) (*Figure 1—figure supplement 2E and F*) indicated no significant difference to foci brightness values (Student's t-test, p=0.67). We also developed a genomically encoded Mig1 reporter using green-red photoswitchable fluorescent protein mEos2 (*McKinney et al., 2009*). Super-resolution stochastic optical reconstruction microscopy (STORM) from hundreds of individual photoactivated tracks indicated the presence of foci, clearly present in nuclei hotspots in live cells at *glucose* (+) (*Figure 1—figure supplement 1*). These results strongly argue that foci formation is not dependent on hypothetical fluorescent protein oligomerization.

We implemented nanoscale tracking based on automated foci detection which combined iterative Gaussian masking and fitting to foci pixel intensity distributions to determine the spatial localization to a lateral precision of 40 nm (*Miller et al., 2015*; *Llorente-Garcia et al., 2014*). Tracking was

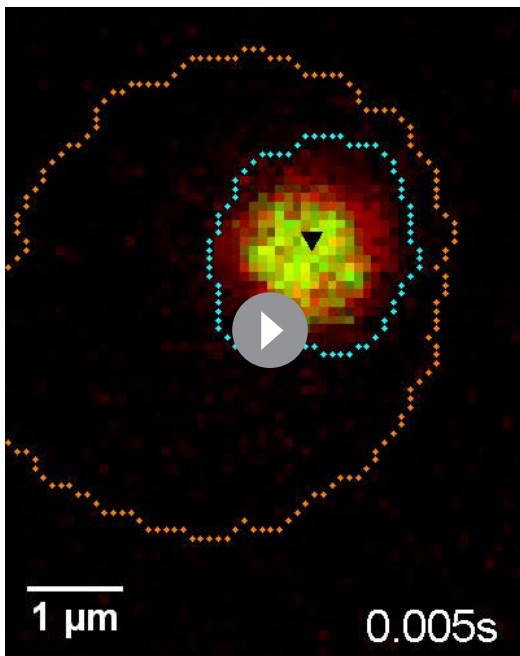

**Video 1.** Dual-color fluorescence microscopy assay at *glucose* (+). Example cell showing *glucose* (+) nuclear Mig1-GFP localization (green, distinct foci black arrows), Nrd1-mCherry (red) and segmented cell body (orange) and nuclear envelope (cyan) indicated, slowed 15x.
DOI: https://doi.org/10.7554/eLife.27451.007

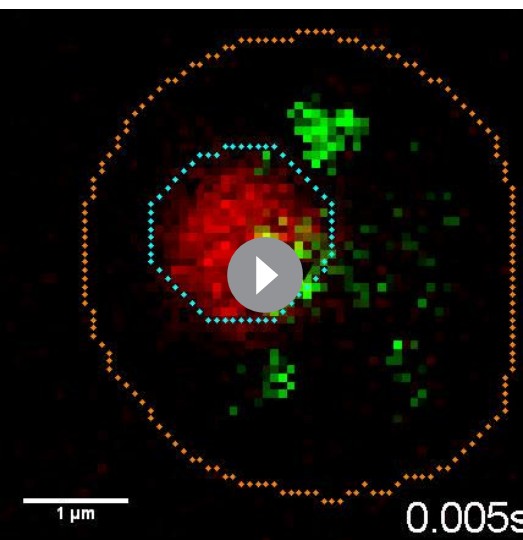

**2.** Dual-color fluorescence microscopy assay at *glucose* (−). Example cell showing *glucose* (−) Mig1-GFP localization (green, distinct foci black arrows), Nrd1-mCherry (red) and segmented cell body (orange) and nuclear envelope (cyan) indicated, slowed 200x.
DOI: https://doi.org/10.7554/eLife.27451.008

coupled to stoichiometry analysis using single GFP photobleaching of foci tracks (*Leake et al., 2006*) and single cell copy number quantification (*Wollman and Leake, 2015*). These methods enabled us to objectively quantify the number of Mig1 molecules associated with each foci, its effective microscopic diffusion coefficient $D$ and spatiotemporal dynamics in regards to its location in the cytoplasm, nucleus or translocating across the nuclear envelope, as well as the copy number of Mig1 molecules associated with each subcellular region and in each cell as a whole. These analyses indicated ~850–1,300 Mig1 total molecules per cell, dependent on extracellular glucose. Quantitative PCR and previous work suggest a higher Mig1 copy number at *glucose* (−) (*Wollman and Leake, 2015*) (*Figure 1D*; *Tables 2* and *3*).

At *glucose* (−) we measured a mean ~950 Mig1 molecules per cell in the cytoplasmic pool (*Figure 1D*) and 30–50 multimeric foci in total per cell, based on interpolating the observed number of foci in the microscope's known depth of field over the entirety of the cell volume. These foci had a mean stoichiometry of 6–9 molecules and mean $D$ of 1–2 $\mu m^2$/s, extending as high as 6 $\mu m^2$/s. In nuclei, the mean foci stoichiometry and $D$ was the same as the cytoplasm to within experimental error (Student's t-test, $p>0.05$, $p=0.99$ and $p=0.83$), with a similar concentration. Trans-nuclear foci, those entering/leaving the nucleus during observed tracking, also had the same mean stoichiometry and $D$ to cytoplasmic values to within experimental error ($p>0.05$, $p=0.60$ and $p=0.79$). However, at *glucose* (+) we measured a considerable increase in the proportion of nuclear foci compared to *glucose* (−), with up to eight foci per nucleus of mean apparent stoichiometry 24–28 molecules, but $D$ lower by a factor of 2, and 0–3 cytoplasmic/trans-nuclear foci per cell (*Figure 2A and B* and *Figure 2—figure Supplement 3*).

## Mig1 cluster localization is dependent on phosphorylation status

To understand how Mig1 clustering was affected by its phosphorylation we deleted the *SNF1* gene which encodes the Mig1-upstream kinase, Snf1, a key regulator of Mig1 phosphorylation. Under Slimfield imaging this strain indicated Mig1 clusters with similar stoichiometry and $D$ as for the wild type strain at *glucose* (+), but with a significant insensitivity to depleting extracellular glucose (*Figure 1—figure supplement 1*, *Figure 2—figure supplement 1A and B*). We also

**Table 2.** Copy number data.
Mean average and SD of copy number in pool and foci in each compartment.

| | Mig1-GFP | | | | Msn2-GFP | | | |
| | *Glucose* (+) | | *Glucose* (−) | | *Glucose* (+) | | *Glucose* (−) | |
| | **Mean** | **SD** | **Mean** | **SD** | **Mean** | **SD** | **Mean** | **SD** |
|---|---|---|---|---|---|---|---|---|
| Cytoplasmic Pool | 509 | 274 | 949 | 394 | 1422 | 977 | 2487 | 1360 |
| Nuclear Pool | 77 | 101 | 140 | 97 | 551 | 608 | 1692 | 1221 |
| Total Pool | 586 | 336 | 1088 | 392 | 1973 | 1585 | 4179 | 2581 |
| Cytoplasmic Spots | 57 | 79 | 311 | 212 | 333 | 196 | 776 | 635 |
| Nuclear Spots | 190 | 99 | 35 | 63 | 81 | 138 | 320 | 269 |
| Total Spots | 246 | 100 | 345 | 203 | 414 | 334 | 1096 | 904 |
| Total Cytoplasm | 580 | 276 | 1156 | 399 | 1755 | 1173 | 3263 | 1995 |
| Total Nuclear | 226 | 155 | 176 | 124 | 632 | 746 | 2012 | 1490 |
| Total Cell | 806 | 353 | 1331 | 352 | 2387 | 1919 | 5274 | 3485 |

DOI: https://doi.org/10.7554/eLife.27451.009

used a yeast strain in which the kinase activity of Snf1 could be controllably inhibited biochemically by addition of cell permeable PP1 analog 1NM-PP1. Slimfield imaging indicated similar results in terms of the presence of Mig1 clusters, their stoichiometry and *D*, but again showing a marked insensitivity towards depleted extracellular glucose indistinguishable from the wild type *glucose* (+) phenotype (*Figure 1—figure supplement 1*, *Figure 2—figure supplement 1C*, *Figure 2—figure supplements 2* and *3* and *Table 4*). We also tested a strain containing Mig1 with four serine phosphorylation sites (Ser222, 278, 311 and 381) mutated to alanine, which were shown to affect Mig1 localization and phosphorylation dependence on extracellular glucose (*DeVit and Johnston, 1999*). Slimfield showed the same pattern of localization as the *SNF1* deletion while retaining the presence of Mig1 clusters (*Figure 2—figure supplement 1D and E*). These results suggest that Mig1 phosphorylation does not affect its ability to form clusters, but does alter their localization bias between nucleus and cytoplasm.

## Cytoplasmic Mig1 is mobile but nuclear Mig1 has mobile and immobile states

The dynamics of Mig1 between cytoplasm and nucleus is critically important to its role in gene regulation. We therefore interrogated tracked foci mobility. We quantified cumulative distribution functions (CDFs) for all nuclear and cytoplasmic tracks (*Gebhardt et al., 2013*). A CDF signifies the probability that foci will move a certain distance from their starting point as a function of time while

**Table 3.** Foci tracking data.
Mean average, SD and mean number detected per cell (N) of stoichiometry values (molecules), and microscopic diffusion coefficients *D* in each compartment detected within the depth of field.

| | Mig1-GFP | | | | | | Msn2-GFP | | | | | |
| | *Glucose* (+) | | | *Glucose* (−) | | | *Glucose* (+) | | | *Glucose* (−) | | |
| | **Mean** | **SD** | **N** | **Mean** | **SD** | **N** | **Mean** | **SD** | **N** | **Mean** | **SD** | **N** |
|---|---|---|---|---|---|---|---|---|---|---|---|---|
| Stoichiometry of Nuclear Spots | 19.0 | 16.2 | 7.2 | 8.5 | 4.8 | 5.8 | 34.5 | 26.6 | 3.5 | 46.5 | 31.6 | 4.7 |
| Diffusion Constant of Nuclear Spots ($\mu m^2$/s) | 0.8 | 0.8 | 7.2 | 1.3 | 1.5 | 5.8 | 0.7 | 0.9 | 3.5 | 0.9 | 0.9 | 4.7 |
| Stoichiometry of Trans-Nuclear Spots | 10.6 | 10.2 | 1.0 | 8.7 | 5.3 | 5.1 | 21.8 | 16.7 | 1.9 | 43.9 | 35.0 | 0.9 |
| Diffusion Constant of Trans-Nuclear Spots ($\mu m^2$/s) | 1.3 | 1.2 | 1.0 | 1.5 | 1.6 | 5.1 | 1.5 | 1.2 | 1.9 | 1.1 | 1.1 | 0.9 |
| Stoichiometry of Cytoplasmic Spots | 6.6 | 4.9 | 1.1 | 7.2 | 3.7 | 17.8 | 25.7 | 19.5 | 4.8 | 30.1 | 17.5 | 4.0 |
| Diffusion Constant of Cytoplasmic Spots ($\mu m^2$/s) | 1.4 | 1.4 | 1.1 | 1.2 | 1.2 | 17.8 | 1.2 | 1.1 | 4.8 | 1.0 | 1.4 | 4.0 |

DOI: https://doi.org/10.7554/eLife.27451.010

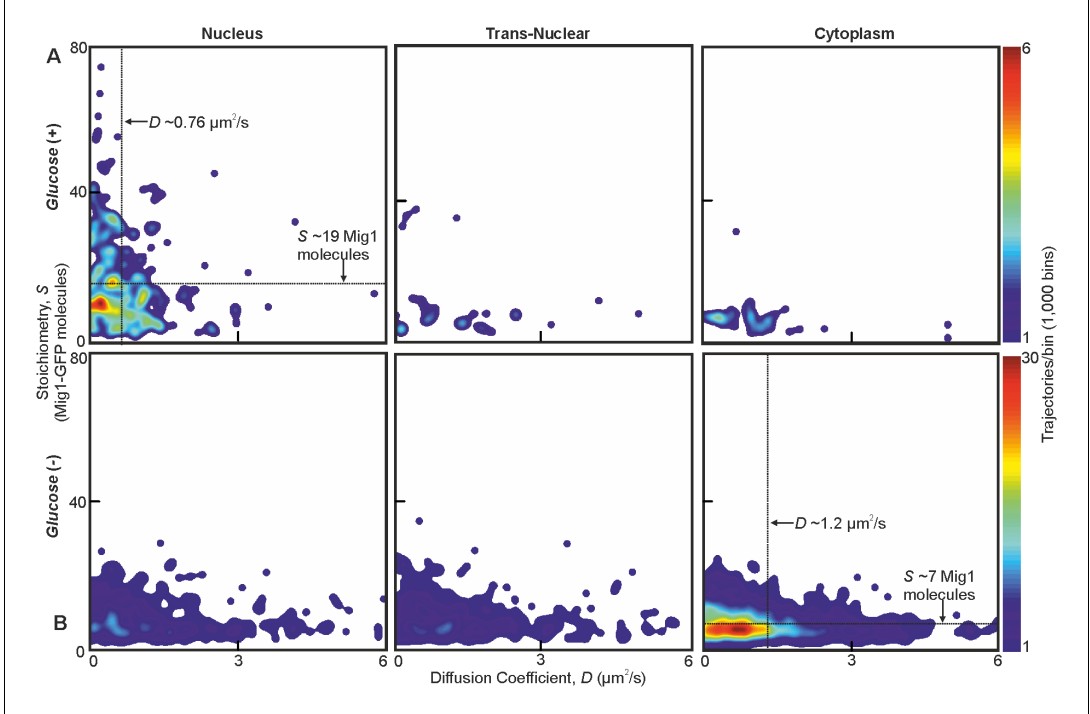

**Figure 2.** Mig1 foci stoichiometry, mobility and localization depend on glucose. Heat map showing dependence of stoichiometry of detected GFP-labeled Mig1 foci with $D$ under (**A**) *glucose* (+) and (**B**) *glucose* (−) extracellular conditions. Mean values for *glucose* (+) nuclear and *glucose* (−) cytoplasmic foci indicated (arrows). n = 30 cells. Heat maps generated using 1000 square pixel grid and 15 pixel width Gaussians at each foci, using variable color scales specified by colorbar on the right.

DOI: https://doi.org/10.7554/eLife.27451.011

The following figure supplements are available for figure 2:

**Figure supplement 1.** Mig1 phosphorylation does not affect clustering but regulates localization.

DOI: https://doi.org/10.7554/eLife.27451.012

**Figure supplement 2.** Wild type Snf1 and analog sensitive have similar effect on Mig1.

DOI: https://doi.org/10.7554/eLife.27451.013

**Figure supplement 3.** Boxplot summary of wild type and mutant Mig1 stoichiometry and microscopic diffusion coefficient.

DOI: https://doi.org/10.7554/eLife.27451.014

tracked. Here, we analyzed only the first displacement of each track to avoid bias toward slowly moving foci (*Gebhardt et al., 2013*). A mixed mobility population can be modeled as the weighted sum of multiple CDFs characterized by different $D$. Cytoplasmic foci at *glucose* (+/-), and nuclear foci at *glucose* (−), were consistent with just a single mobile population (*Figure 3—figure supplement 1*) whose $D$ of 1–2 $\mu m^2/s$ was consistent with earlier observations. However, nuclear foci at *glucose* (+) indicated a mixture of mobile and immobile components (*Figure 3A*). These results, substantiated by fitting two Gamma functions to the distribution of estimated $D$ (*Stracy et al., 2015*) for *glucose* (+) nuclear foci (*Figure 3A*, inset), indicate 20–30% of nuclear foci are immobile, consistent with a DNA-bound state. Mean square displacement analysis of foci tracks sorted by stoichiometry indicated Brownian diffusion over short timescales of a few tens of ms but increasingly anomalous diffusion over longer timescales > 30 ms (*Figure 3B*). These results are consistent with *glucose* (+) Mig1 diffusion being impacted by interactions with nuclear structures, similar to that reported for other transcription factors (*Izeddin et al., 2014*). Here however this interaction is dependent on extracellular glucose despite Mig1 requiring a pathway of proteins to detect it, unlike the more direct detection mechanism of the prokaryotic *lac* repressor. A strain in which mCherry labeled Mig1 had its Zn finger deleted (Δaa36-91) (*Lin et al., 2015*) indicated no significant immobile cluster population at *glucose* (+/-) with CDF analysis (*Figure 3—figure supplement 1*). We conclude that Mig1 clusters bind with a relatively high association constant to the DNA via their Zn finger motif with direct glucose dependence.

**Table 4.** *snf1Δ* foci tracking and copy number data.

Upper panel: Mean average, SD and mean number detected per cell (N) of stoichiometry values (molecules), and microscopic diffusion coefficients *D* in each compartment detected within the depth of field. Lower panel: Mean average and SD of copy number in pool and foci in each compartment.

| | Mig1-GFP *snf1Δ* | | | | | |
| --- | --- | --- | --- | --- | --- | --- |
| | *Glucose* (+) | | | *Glucose* (−) | | |
| | **Mean** | **SD** | **N** | **Mean** | **SD** | **N** |
| Stoichiometry of Nuclear Spots | 17.5 | 10.9 | 13.2 | 23.5 | 15.4 | 10.9 |
| Diffusion Constant of Nuclear Spots ($\mu m^2/s$) | 1.1 | 1.1 | 13.2 | 0.7 | 0.8 | 10.9 |
| Stoichiometry of Trans-Nuclear Spots | 8.9 | 6.0 | 1.2 | 12.7 | 6.1 | 0.5 |
| Diffusion Constant of Trans-Nuclear Spots ($\mu m^2/s$) | 1.9 | 2.0 | 1.2 | 1.1 | 1.4 | 0.5 |
| Stoichiometry of Cytoplasmic Spots | 6.2 | 2.2 | 5.0 | 8.3 | 4.1 | 9.1 |
| Diffusion Constant of Cytoplasmic Spots ($\mu m^2/s$) | 1.3 | 1.2 | 5.0 | 1.0 | 1.2 | 9.1 |
| Copy Numbers | | | | | | |
| Cytoplasmic Pool | 947 | 728 | 30 | 608 | 450 | 30 |
| Nuclear Pool | 807 | 398 | 30 | 611 | 325 | 30 |
| Total Pool | 1754 | 1127 | 30 | 1219 | 775 | 30 |
| Cytoplasmic Spots | 118 | 169 | 30 | 334 | 374 | 30 |
| Nuclear Spots | 162 | 69 | 30 | 164 | 71 | 30 |
| Total Spots | 280 | 238 | 30 | 498 | 445 | 30 |
| Total Cytoplasm | 1065 | 897 | 30 | 941 | 824 | 30 |
| Total Nuclear | 969 | 467 | 30 | 775 | 396 | 30 |
| Total Cell | 2034 | 1364 | 30 | 1717 | 1220 | 30 |

DOI: https://doi.org/10.7554/eLife.27451.015

## Mig1 nuclear translocation selectivity does not depend on glucose but is mediated by interactions away from the nuclear envelope

Due to the marked localization of Mig1 towards nucleus/cytoplasm at *glucose* (+/-) respectively, we asked whether this spatial bias was due to selectivity initiated during translocation at the nuclear envelope. By converting trans-nuclear tracks into coordinates parallel and perpendicular to the measured nuclear envelope position, and synchronizing origins to be the nuclear envelope crossing point, we could compare spatiotemporal dynamics of different Mig1 clusters during translocation. A heat map of spatial distributions of translocating clusters indicated a hotspot of comparable volume to that of structures of budding yeast nuclear pore complexes (*Adam, 2001*) and accessory nuclear structures of cytoplasmic nucleoporin filaments and nuclear basket (*Strambio-De-Castillia et al., 2010*), with some nuclear impairment to mobility consistent with restrained mobility (*Figure 3C*). We observed a dwell in cluster translocation across the 30–40 nm width of the nuclear envelope (*Figure 3D*). At *glucose* (+) the proportion of detected trans-nuclear foci was significantly higher compared to *glucose* (−), consistent with Mig1's role to repress genes. The distribution of dwell times could be fitted using a single exponential function with ~10 ms time constant similar to previous estimates for transport factors (*Yang et al., 2004*). However, although the relative proportion of trans-nuclear foci was much lower at *glucose* (−) compared to *glucose* (+), the dwell time constant was found to be insensitive to glucose (*Figure 3E*). This insensitivity to extracellular chemical signal demonstrates, surprisingly, that there is no direct selectivity on the basis of transcription factor phosphorylation state by nuclear pore complexes themselves, suggesting that cargo selectivity mechanisms of nuclear transport (*Lowe et al., 2010*), as reported for a range of substrates, is blind to the phosphorylation state. Coupled with our observation that Mig1 at *glucose* (−) does not exhibit significant immobility in the nucleus and that Mig1 lacking the Zn finger still accumulates in the nucleus at *glucose* (+) (*Figure 1—figure supplement 1*), this suggests that Mig1 localization is driven by

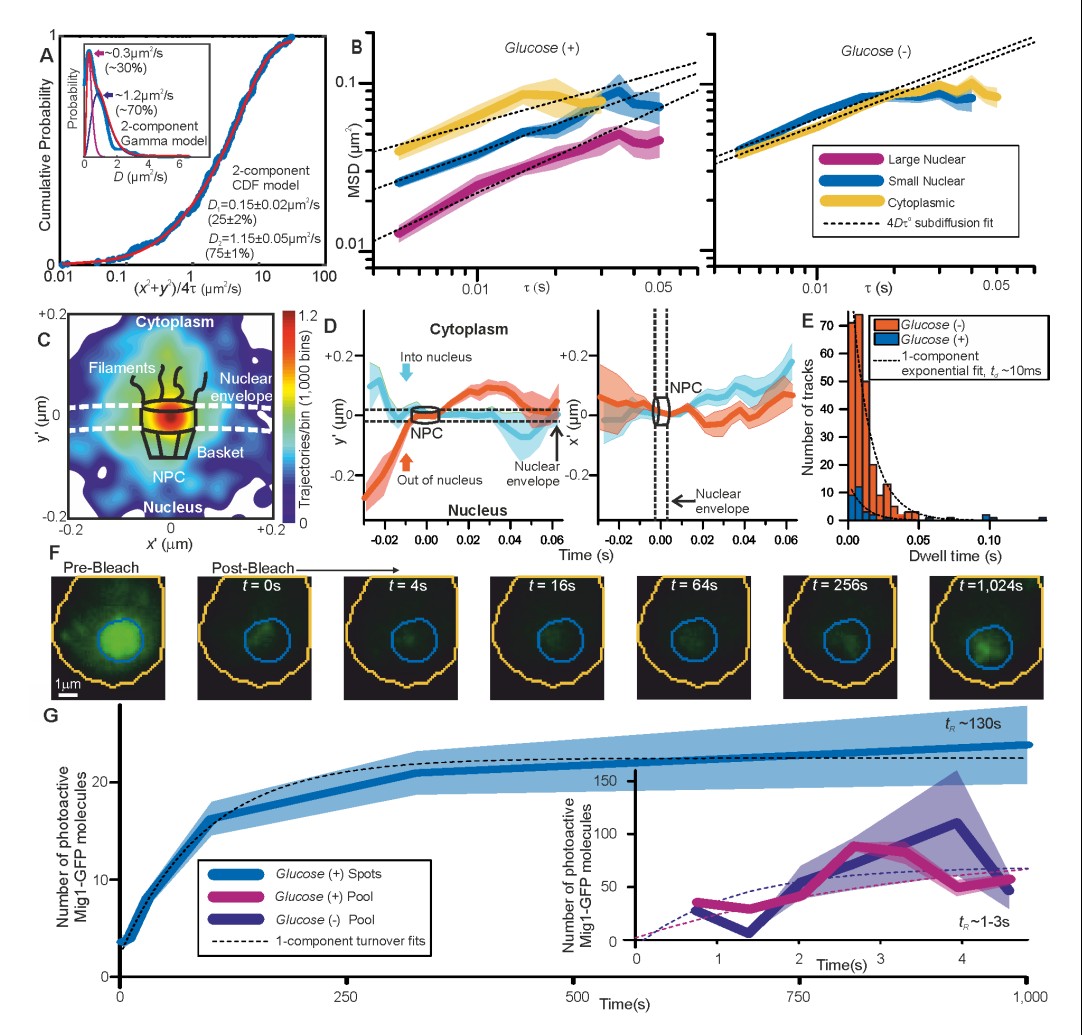

**Figure 3.** Repressor clusters have heterogeneous mobility depending on localization. (**A**) Cumulative probability, *glucose* (+) nuclear tracks (blue) and two component exponential fit (red), with dual Gamma fit to *D* (inset) with similar parameters. (**B**) Mean MSD *vs* τ (i.e. time interval tau) from cytoplasmic (yellow), small (blue, stoichiometry ≤ 20 Mig1-GFP molecules) and large nuclear (purple, stoichiometry > 20 Mig1-GFP molecules) foci, SEM indicated, on log-log axes, n = 30 cells for *glucose* (+) and (−). Anomalous diffusion model fits to time intervals ≤ 30 ms (dashed black line), anomalous coefficient α = 0.4–0.8. (**C**) Heat map of trans-nuclear track localizations normalized to crossover point, generated using 1000 square pixel grid and 10 pixel width Gaussians at each localization (**D**) distance parallel (left) and perpendicular (right) to nuclear envelope with time, normalized to crossover point for Mig1-GFP foci entering (blue) and leaving the nucleus (red), (**E**) dwell times at nuclear envelope and single exponential fits (dotted). (**F**) Example *glucose* (+) single cell FRAP Slimfield images, fixed display scale (**G**) mean and SEM nuclear intensity after bleaching, n = 5 and 7 cells for *glucose* (-/+), respectively.

DOI: https://doi.org/10.7554/eLife.27451.016

The following figure supplement is available for figure 3:

**Figure supplement 1.** Cumulative probability distance analysis reveals a single mobile population in the cytoplasm at glucose (+/-) and in the nucleus and glucose (−).

DOI: https://doi.org/10.7554/eLife.27451.017

changes in Mig1 binding affinity to other proteins, within for example the general corepressor complex (*Treitel and Carlson, 1995*), or outside the nucleus not involving the nuclear pore complex.

## Mig1 nuclear foci bound to targets turn over slowly as whole clusters of ~ 7–9 molecules in >100 s

To further understand the mechanisms of Mig1 binding/release during gene regulation we sought to quantify kinetics of these events at Mig1 targets. By modifying our microscope we could implement

an independent focused laser path using the same laser source, enabling us to use fluorescence recovery after photobleaching (FRAP) to probe nuclear Mig1 turnover. The focused laser rapidly photobleached GFP content in cell nuclei in <200 ms (*Figure 3F*). We could then monitor recovery of any fluorescence intensity by illuminating with millisecond Slimfield stroboscopically as opposed to continuously to extend the observation timescale to >1,000 s. Using automated foci detection we could separate nuclear pool and foci content at each time point for each cell. These analyses demonstrated measurable fluorescence recovery for both components, which could be fitted by single exponentials indicating fast recovery of pool at both *glucose* (−) and (+) with a time constant <5 s but a larger time constant at *glucose* (+) for nuclear foci > 100 s (*Figure 3G*). Further analysis of intensity levels at each time point revealed a stoichiometry periodicity in nuclear foci recovery equivalent to 7–9 GFP molecules (*Figure 4—figure supplement 1A*), but no obvious periodicity in stoichiometry measurable from pool recovery. An identical periodicity within experimental error was measured from nuclear foci at *glucose* (+) in steady-state (*Figure 4A*). These periodicity values in Mig1 stoichiometry were consistent with earlier observations for cytoplasmic and trans-nuclear clusters at *glucose* (+/-), and in the nucleus at *glucose* (−), with mean stoichiometry ~7 molecules. These data taken as a whole clearly suggest that molecular turnover at nuclear foci of Mig1 bound to its target genes occurs in units of single clusters, as opposed to single Mig1 monomers.

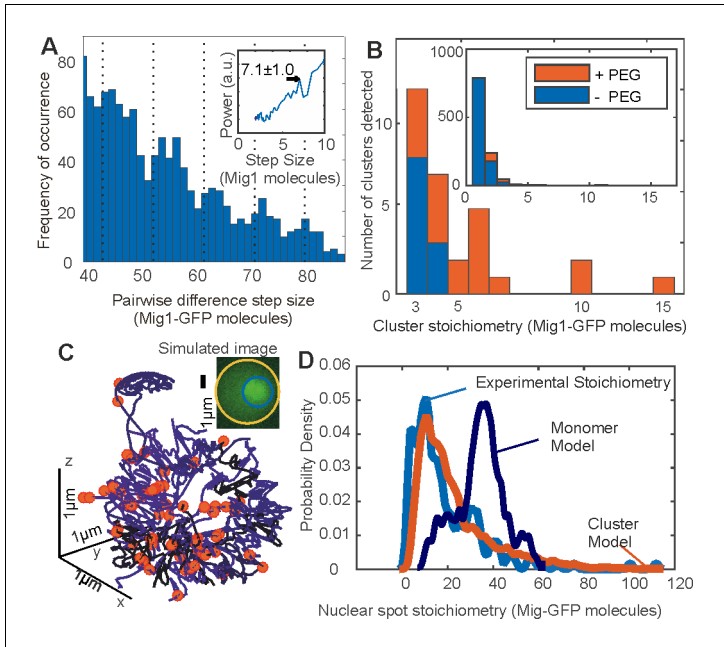

**Figure 4.** Mig1 clusters are stabilized by depletion forces and bind to promoter targets. (**A**) Zoom-in on pairwise difference distribution for stoichiometry of Mig1-GFP foci, 7-mer intervals (dashed) and power spectrum (inset), mean and Gaussian sigma error (arrow). (**B**) Stoichiometry for Mig1-GFP clusters in vitro in PEG absence (blue)/ presence (red). n = 1000 foci. Inset shows the full range while outer zooms in on cluster stoichiometry. (**C**) 3C model of chromosomal DNA (blue shaded differently for each chromosome) with overlaid Mig1 promoter binding sites from bioinformatics (red), simulated image based on model with realistic signal and noise added (inset). (**D**) Cluster (red) and monomer (dark blue) model (goodness-of-fit $R^2$ < 0) for Mig1-GFP stoichiometry (10 replicates) compared against experimental data (cyan, $R^2$ = 0.75).

DOI: https://doi.org/10.7554/eLife.27451.018

The following figure supplements are available for figure 4:

**Figure supplement 1.** Additional Mig1 cluster investigations.
DOI: https://doi.org/10.7554/eLife.27451.019

**Figure supplement 2.** In vitro cluster characterization.
DOI: https://doi.org/10.7554/eLife.27451.020

**Figure supplement 3.** Additional 3C modelling.
DOI: https://doi.org/10.7554/eLife.27451.021

## Mig1 clusters are spherical, a few tens of nm wide

Our observations from stoichiometry, dynamics and kinetics, which supported the hypothesis that functional clusters of Mig1 perform the role of gene regulation, also suggested an obvious prediction in terms of the size of observed foci: the physical diameter of a multimeric cluster should be larger than that of a single Mig1 monomer. We therefore sought to quantify foci widths from Slimfield data by performing intensity profile analysis on background-corrected pixel values over each foci image. The diameter was estimated from the measured width corrected for motion blur due to particle diffusion in the sampling time of a single image frame, minus that measured from single purified GFP molecules immobilized to the coverslip surface in separate in vitro experiments. This analysis revealed diameters of 15–50 nm at *glucose* (−), which showed an increase with foci stoichiometry S that could be fitted with a power law dependence $S^a$ (*Figure 4—figure supplement 1B*) with optimized exponent *a* of 0.32 ± 0.06 (±SEM). Immuno-gold electron microscopy of fixed cells probed with anti-GFP antibody confirmed the presence of GFP in 90 nm cryosections with some evidence of clusters containing up to 7 Mig1 molecules (*Figure 4—figure supplement 1C*), however, the overall labeling efficiency was relatively low with sparse labelling in the nucleus in particular, possibly as a consequence of probe inaccessibility, resulting in relatively poor statistics. A heuristic tight packing model for GFP labeled Mig1 monomers in each cluster predicts that, in the instance of an idealized spherical cluster, *a* = 1/3. Our data at *glucose* (−) thus supports the hypothesis that Mig1 clusters have a spherical shape. For nuclear foci at *glucose* (+) we measured larger apparent diameters and stoichiometries, consistent with >1 individual Mig1 cluster being separated by less than our measured ~200 nm optical resolution limit. This observation agrees with earlier measurements of stoichiometry periodicity for nuclear foci at *glucose* (+). In other words, that higher apparent stoichiometry nuclear foci are consistent with multiple individual Mig1 clusters each containing ~7 molecules separated by a nearest neighbor distance <200 nm and so detected as a single fluorescent foci.

## Clusters are stabilized by depletion forces

Since we observed Mig1 clusters in live cells using Slimfield imaging we wondered if these could be detected and further quantified using other methods. However, native gel electrophoresis on extracts from Mig1-GFP cells (*Figure 4—figure supplement 2A*) indicated a single stained band for Mig1, which was consistent with denaturing SDS-PAGE combined with western blotting using recombinant Mig1-GFP, and protein extracts from the parental cells which included no fluorescent reporter (*Figure 4—figure supplement 2B and C*). Slimfield imaging on purified Mig1-GFP in vitro under identical imaging conditions for live cells similarly indicated monomeric Mig1-GFP foci in addition to a small fraction of brighter foci which were consistent with predicted random overlap of monomer images. However, on addition of low molecular weight polyethylene glycol (PEG) at a concentration known to mimic small molecule 'depletion' forces in live cells (*Phillip and Schreiber, 2013*) we detected significant numbers of multimeric foci (*Figure 4B* and *Figure 4—figure Supplement 2D*). Depletion is an entropic derived attractive force which results from osmotic pressure between particles suspended in solution that are separated by distances short enough to exclude other surrounding smaller particles. Purified GFP alone under identical conditions showed no such effect (*Figure 4—figure supplement 2E*). These results support a hypothesis that clusters are present in live cells regardless of the concentration of extracellular glucose, which are stabilized by depletion components that are lost during biochemical purification.

## Chromosome structure modeling supports a cluster binding hypothesis

We speculated that Mig1 cluster-mediated gene regulation had testable predictions in regards to the nuclear location of Mig1 at elevated extracellular glucose. We therefore developed quantitative models to simulate the appearance of realistic images of genome-bound Mig1-GFP at *glucose* (+). We used sequence analysis to infer locations of Mig1 binding sites in the yeast genome, based on alignment matches to previously identified 17 bp Mig1 target patterns (*Lundin et al., 1994*) which comprised conserved AT-rich 5 bp and GC-rich 6 bp sequences. In scanning the entire *S. cerevisiae* genome we found >3000 hits though only 112 matches for likely gene regulatory sites located in promoter regions (*Table 5*). We mapped these candidate binding sites onto specific 3D locations (*Figure 4C*) obtained from a consensus structure for budding yeast chromosomes based on 3C data

**Table 5.** Number of potential Mig1 target promoter sites per chromosome.
List of *S.cerevisiae* chromosomes indicating the length of a chromosome, total number of potential Mig1 target sites identified and then the number of sites on promoters assuming a promoter region up to 500 bp upstream of a gene.

| Chromosome | Length (bp) | N sites identified | N promoter sites |
| --- | --- | --- | --- |
| I | 230218 | 41 | 1 |
| II | 813184 | 134 | 10 |
| III | 316620 | 52 | 2 |
| IV | 1531933 | 240 | 14 |
| V | 576874 | 109 | 8 |
| VI | 270161 | 58 | 4 |
| VII | 1090940 | 168 | 13 |
| VIII | 562643 | 92 | 2 |
| IX | 439888 | 94 | 8 |
| X | 745751 | 125 | 6 |
| XI | 666816 | 117 | 6 |
| XII | 1078177 | 194 | 12 |
| XIII | 924431 | 157 | 6 |
| XIV | 784333 | 135 | 3 |
| XV | 1091291 | 185 | 11 |
| XVI | 948066 | 163 | 6 |

DOI: https://doi.org/10.7554/eLife.27451.022

(*Duan et al., 2010*). We generated simulated images, adding experimentally realistic levels of signal and noise, and ran these synthetic data through the same tracking software as for experimental data. We used identical algorithm parameters throughout and compared these predictions to the measured experimental stoichiometry distributions.

In the first instance we used these locations as coordinates for Mig1 monomer binding, assuming that just a single Mig1 molecule binds to a target. Copy number analysis of Slimfield data (*Table 2*) indicated a mean ~190 Mig1 molecules per cell associated with nuclear foci, greater than the number of Mig1 binding sites in promoter regions. We assigned 112 molecules to target promoter binding sites, then assigned the remaining 78 molecules randomly to non-specific DNA coordinates of the chromosomal structure. We included the effects of different orientations of the chromosomal structure relative to the camera by generating simulations from different projections and included these in compiled synthetic datasets.

We then contrasted monomer binding to a cluster binding model, which assumed that a whole cluster comprising 7 GFP labeled Mig1 molecules binds a single Mig1 target. Here we randomly assigned the 190 Mig1 molecules into just 27 (i.e. ~190/7) 7-mer clusters to the set of 112 Mig1 target promoter sites. We also implemented improvements of both monomer and cluster binding models to account for the presence of trans-nuclear tracks. Extrapolating the number of detected trans-nuclear foci in our microscope's depth of field over the whole nuclear surface area indicated a total of ~130 Mig1 molecules at *glucose* (+) inside the nucleus prior to export across the cytoplasm. We simulated the presence of these trans-nuclear molecules either using 130 GFP-labeled Mig1 molecules as monomers, or as 18 (i.e. ~130/7) 7-mer clusters at random 3D coordinates over the nuclear envelope surface (*Figure 4—figure supplement 3*).

We discovered that a cluster binding model which included the presence of trans-nuclear foci generated excellent agreement to the experimental foci stoichiometry distribution ($R^2 = 0.75$) compared to a very poor fit for a monomer binding model ($R^2 < 0$) (*Figure 4D*). The optimized cluster model fit involved on average ~25% of promoter loci to be bound across a population of simulated cells by a 7-mer cluster with the remaining clusters located non-specifically, near the nuclear envelope, consistent with nuclear transit. This structural model supports the hypothesis that the functional

unit of Mig1-mediated gene regulation is a cluster of Mig1 molecules, as opposed to Mig1 acting as a monomer.

## The activator Msn2 also forms functional clusters

We wondered if the discovery of transcription factor clusters was unique to specific properties of the Mig1 repressor, as opposed to being a more general feature of other Zn finger transcription factors. To address this question we prepared a genomically encoded GFP fusion construct of a similar protein Msn2. Nrd1-mCherry was again used as a nuclear marker (*Figure 1—figure supplement 1*). Msn2 acts as an activator and not a repressor, which co-regulates several Mig1 target genes but with the opposite nuclear localization response to glucose (*Lin et al., 2015*). On performing Slimfield under identical conditions to the Mig1-GFP strain we again observed a significant population of fluorescent Msn2 foci, which had comparable $D$ and stoichiometry to those estimated earlier for Mig1 (*Table 2*). The key difference with the data from the Mig1-GFP strain was that Msn2, unlike Mig1, demonstrated high apparent foci stoichiometry values and lower values of $D$ at *glucose* (−), which was consistent with its role as an activator of the same target genes as opposed to a repressor (*Figure 5A and B*). Immuno-gold electron microscopy of fixed Msn2-GFP cells confirmed the presence of GFP in 90 nm cryosections with evidence for clusters of comparable diameters to Mig1-GFP (*Figure 4—figure supplement 1C*), but with the same technical caveats and poor statistics as reported for the Mig1-GFP dataset. These results support the hypothesis that two different eukaryotic transcription factors that have antagonist effects on the same target genes operate as molecular clusters.

To test the functional relevance of Mig1 and Msn2 clusters we performed Slimfield on a strain in which Mig1 and Msn2 were genomically labeled using mCherry and orange fluorescent protein mKO2, respectively (*Lin et al., 2015*). This strain also contained a plasmid with GFP labeled PP7 protein to report on nuclear mRNA expressed specifically from the glycogen synthase *GSY1* gene, whose expression can be induced by glucose starvation and is a target of Mig1 and Msn2, labelled with 24 repeats of the PP7 binding sequence (*Unnikrishnan et al., 2003*). In switching from *glucose* (+) to (−) and observing the same cell throughout, we measured PP7 accumulating with similar localization patterns to those of Mig1 clusters at *glucose* (+) (*Figure 5C*). No accumulation was observed with the mutant Mig1 lacking the Zn finger, in line with previous observations (*Lin et al., 2015*). We calculated the numerical overlap integral between these Mig1 and PP7 foci (*Figure 5D*), indicating a high mean of ~0.95, where one is the theoretical maximum for 100% colocalization in the absence of noise (*Llorente-Garcia et al., 2014*). We also observed similar high colocalization between Msn2-mKO2 clusters and PP7-GFP at *glucose* (−) (*Figure 5E*). These results demonstrate a functional link between the localization of Mig1 and Msn2 clusters, and the transcribed mRNA from their target genes.

## Mig1 and Msn2 possess intrinsic disorder which may favor clustering

Since both Mig1 and Msn2 demonstrate significant populations of clustered molecules in functional cell strains we asked the question if there were features common to the sequences of both proteins which might explain this behavior. To address this question we used multiple sequence alignment to determine conserved structural features of both proteins, and secondary structure prediction tools with disorder prediction algorithms. As expected, sequence alignment indicated the presence of the Zn finger motif in both proteins, with secondary structure predictions suggesting relatively elongated structures (*Figure 6A*). However, disorder predictions indicated multiple extended intrinsically disordered regions in both Mig1 and Msn2 sequences with an overall proportion of disordered content >50%, as high as 75% for Mig1 (*Figure 6B*; *Table 6*). We measured a trend from a more structured region of Mig1 towards the N-terminus and more disordered regions towards the C-terminus. Msn2 demonstrated a similar bipolar trend but with the structured Zn finger motif towards the C-terminus and the disordered sequences towards the N-terminus. We then ran the same analysis as a comparison against the prokaryotic transcription factor LacI, which represses expression from genes of the *lac* operon as part of the prokaryotic glucose sensing pathway. The predicted disorder content in the case of LacI was <50%. In addition, further sequence alignment analysis predicted that at least 50% of candidate phosphorylation sites in either Mig1 or Msn2 lie within these intrinsically disordered sequences (*Table 6*; *Figure 6A*). An important observation reported previously is that the

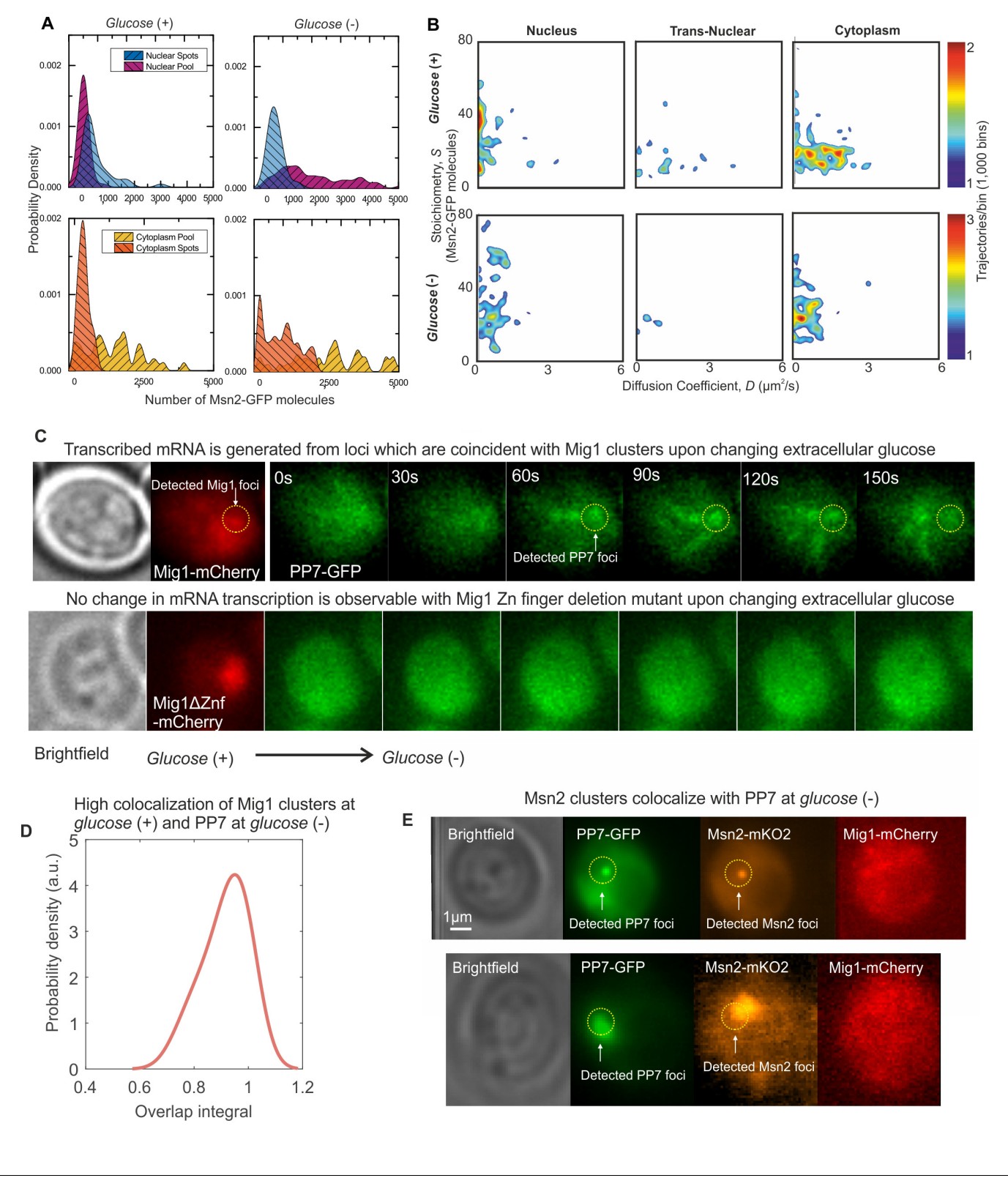

**Figure 5.** Msn2 and Mig1 forms functional clusters colocalized to transcribed mRNA from their target genes. (**A**) Kernel density estimations for Msn2-GFP in pool and foci for cytoplasm and nucleus at *glucose* (+/-). (**B**) Heat maps showing dependence of stoichiometry and *D* of detected Msn2-GFP foci, n = 30 cells. (**C**) Slimfield imaging on the same cell in which microfluidics is used to switch from *glucose* (+) to *glucose* (−) indicating the emergence of PP7-GFP foci at *glucose* (−) which are coincident with Mig1-mCherry foci at *glucose* (+), dependent on the Mig1 Zn finger (same intensity

*Figure 5 continued*

display scales throughout). These Mig1 and PP7 foci have a high level of colocalization as seen from (**D**) the distribution of the numerical overlap integral between foci in red and green channels at *glucose* (+) and *glucose* (−) respectively, peaking at ~0.95. n = 21 cells. (**E**) Two example cells showing at *glucose* (−) Msn2-mKO2 foci colocalize with PP7-GFP foci. PP7-2xGFP and Msn2-mKO2 images are frame averages of ~1000 frames, Mig1-mCherry is a Slimfield image.

DOI: https://doi.org/10.7554/eLife.27451.023

comparatively highly structured LacI exhibits no obvious clustering behavior from similar high-speed fluorescence microscopy tracking on live bacteria (*Mahmutovic et al., 2015*). Intrinsically disordered proteins are known to undergo phase transitions which may enable cluster formation and increase the likelihood of binding to nucleic acids (*Uversky et al., 2015*; *Toretsky and Wright, 2014*). It has been shown that homo-oligomerization is energetically more favorable than hetero-oligomerization (*Goodsell and Olson, 2000*). Moreover, symmetrical arrangement of the same protein can increase accessibility of the protein to binding partners, generate new binding sites, or increase complex specificity and diversity in general (*Fong et al., 2009*). We measured significant changes in circular dichroism of the Mig1 fusion construct upon addition of PEG in the wavelength range 200–230 nm (*Figure 6C*) known to be sensitive to transitions between ordered and intrinsically disordered states (*Sode et al., 2006*; *Avitabile et al., 2014*). Since the Zn finger motif lies towards the opposite terminus to the disordered content for both Mig1 and Msn2 this may suggest a molecular bipolarity which could stabilize a cluster core while exposing Zn fingers on the surface enabling interaction with accessible DNA. This structural mechanism has analogies to that of phospholipid interactions driving micelle formation, however mediated here through disordered sequence interactions as opposed to hydrophobic forces (*Figure 6C*). The prevalence of phosphorylation sites located in disordered regions may also suggest a role in mediating affinity to target genes, similar to protein-protein binding by phosphorylation and intrinsic disorder coupling (*Nishi et al., 2013*).

## Discussion

Our findings address a totally underexplored and novel aspect of gene regulation with technology that has not been available until recently. In summary, we observe that the repressor protein Mig1 forms clusters which, upon extracellular glucose detection, localize dynamically from the cytoplasm to bind to locations consistent with promoter sequences of its target genes. Similar localization events were observed for the activator Msn2 under glucose limiting conditions. Moreover, Mig1 and Msn2 oligomers colocalized with mRNA transcribed from *GSY1* gene at glucose (+/-), respectively. Our results therefore strongly support a functional link between Mig1 and Msn2 transcription factor clusters and target gene expression. The physiological role of multivalent transcription factor clusters has been elucidated through simulations (*Schmidt et al., 2014*) but unobserved until now. These simulations show that intersegmental transfer between sections of nuclear DNA was essential for factors to find their binding sites within physiologically relevant timescales and requires multivalency. Previous single-molecule studies of p53 (*Mazza et al., 2012*) and TetR (*Normanno et al., 2015*) in human cancer cells have also suggested a role for non-specific (i.e. sequence independent) transcription factor searching along the DNA. Our findings address the longstanding question of how transcription factors find their targets in the genome so efficiently. Evidence for higher molecular weight Mig1 states from biochemical studies has been suggested previously (*Needham and Trumbly, 2006*). A Mig1-His-HA construct was overexpressed in yeast and cell extracts run in different glucose concentrations through sucrose density centrifugation. In western blots, a higher molecular weight band was observed, attributed to a hypothetical cofactor protein. However, no cofactor was detected and none reported to date. The modal molecular weight observed was ~four times that of Mig1 but with a wide observed distribution consistent with our mean detected cluster size of ~7 molecules. The authors only reported detecting higher molecular weight states in the nucleus in repressing conditions.

Clustering of nuclear factors has been reported previously in other systems using single-molecule techniques. In particular, RNA polymerase clustering in the nucleus has been shown to have a functional role in gene regulation through putative transcription factories (*Cisse et al., 2013*; *Cho et al., 2016*). Other nuclear protein clusters have been shown to have a functional role (*Qian et al., 2014*)

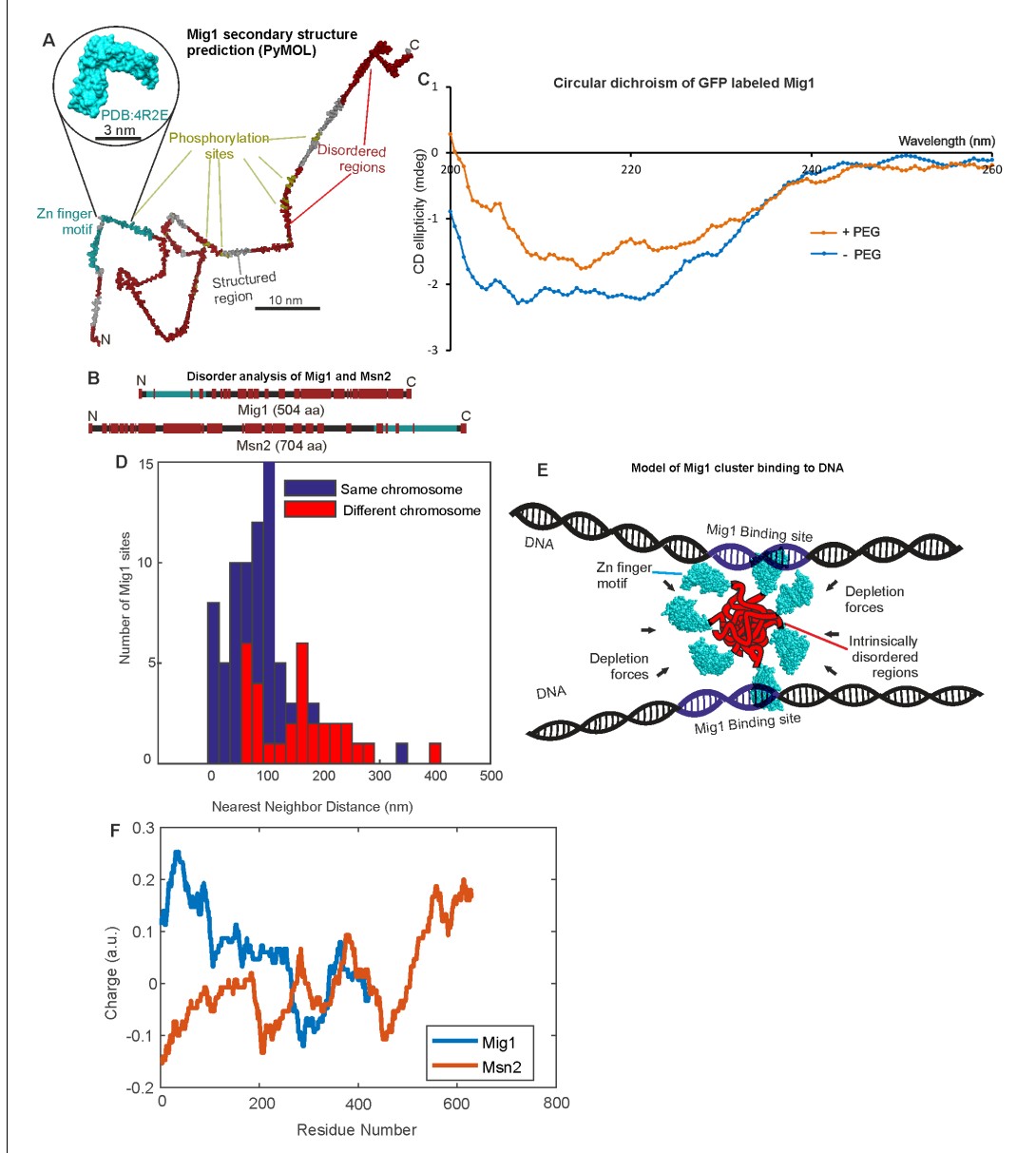

**Figure 6.** Mig1 and Msn2 contain disordered sequences which may mediate cluster formation. (A) Structural prediction for Mig1; Zn finger motif (cyan), disordered sections (red) from PyMOL, beta sheet (gray), phosphorylation sites (yellow); zoom-in indicates structure of conserved Zn finger from PSI-BLAST to PDB ID: 4R2E (Wilms tumor protein, WT1). (B) DISOPRED prediction for Mig1 and Msn2; disordered regions (red), Zn finger regions (cyan). (C) Circular dichroism of Mig1-GFP in vitro in PEG absence (blue)/presence (orange) (D) Distribution of nearest neighbor distances for Mig1 sites within promoters on same (blue) or different (red) chromosome. (E) Schematic of depletion-stabilized Mig1 cluster bound to multiple promoter targets (Zn finger PDB ID: 4R2E). (F) Amino acid residue electrostatic charge plots for Mig1 and Msn2 from EMBOSS (*Rice et al., 2000*) Residues 'D' and 'E' are assigned a charge of −1, 'K' and 'R' a charge of + 1, and the residue 'H' is assigned a charge of + 0.5, then a rolling 75 amino acid residue window is used. Figures and Tables.

DOI: https://doi.org/10.7554/eLife.27451.024

and the Bicoid transcription factor in *Drosophila melanogaster* embryos has been shown to form clusters partially mediated by regions of intrinsic disorder (*Mir, 2017*).

Our measured turnover of genome-bound Mig1 has similar timescales to that estimated for nucleoid-bound LacI (*Mahmutovic et al., 2015*), but similar rates of turnover have also been observed in yeast for a DNA-bound activator (*Karpova et al., 2008*). Faster off rates have been observed during single particle tracking of the DNA-bound fraction of the glucocorticoid receptor (GR) transcription

**Table 6.** Bioinformatics analysis for intrinsically disordered sequences.
Predictions for the presence of intrinsically disordered sequences in Mig1, Msn2 and LacI, and of the positions of phosphorylation sites in Mig1 and Msn2.

**Msn2:**

| | |
|---|---|
| Predicted residues: 704 | Number Disordered Regions: 12 |
| Number residues disordered: 394 | Longest Disordered Region:145 |
| Overall percent disordered: 55.97 | Average Prediction Score: 0.5577 |
| Predicted disorder segment [1 - 2] | Average Strength = 0.8759 |
| Predicted disorder segment [16 - 33] | Average Strength = 0.6958 |
| Predicted disorder segment [55 - 199] | Average Strength = 0.8311 |
| Predicted disorder segment [222 - 249] | Average Strength = 0.8237 |
| Predicted disorder segment [322 - 365] | Average Strength = 0.8820 |
| Predicted disorder segment [410 - 428] | Average Strength = 0.7475 |
| Predicted disorder segment [469 - 480] | Average Strength = 0.6545 |
| Predicted disorder segment [510 - 549] | Average Strength = 0.8040 |
| Predicted disorder segment [572 - 641] | Average Strength = 0.9319 |
| Predicted disorder segment [660 - 667] | Average Strength = 0.6829 |
| Predicted disorder segment [694 - 695] | Average Strength = 0.5325 |
| Predicted disorder segment [699 - 704] | Average Strength = 0.6783 |

**Mig1:**

| | |
|---|---|
| Predicted residues: 504 | Number Disordered Regions: 9 |
| Number residues disordered: 372 | Longest Disordered Region: 95 |
| Overall percent disordered: 73.81 | Average Prediction Score: 0.7008 |
| Predicted disorder segment [1 - 12] | Average Strength = 0.8252 |
| Predicted disorder segment [25 - 33] | Average Strength = 0.6502 |
| Predicted disorder segment [77 - 171] | Average Strength = 0.8758 |
| Predicted disorder segment [173 - 240] | Average Strength = 0.9051 |
| Predicted disorder segment [242 - 249] | Average Strength = 0.5554 |
| Predicted disorder segment [254 - 272] | Average Strength = 0.7890 |
| Predicted disorder segment [292 - 310] | Average Strength = 0.8225 |
| Predicted disorder segment [327 - 386] | Average Strength = 0.8355 |
| Predicted disorder segment [423 - 504] | Average Strength = 0.9136 |

**LacI:**

| | |
|---|---|
| Predicted residues: 360 | Number Disordered Regions: 8 |
| Number residues disordered: 149 | Longest Disordered Region: 48 |
| Overall percent disordered: 41.39 | Average Prediction Score: 0.4418 |
| Predicted disorder segment [1 - 4] | Average Strength = 0.6245 |
| Predicted disorder segment [18 - 52] | Average Strength = 0.6710 |
| Predicted disorder segment [55 - 81] | Average Strength = 0.7443 |
| Predicted disorder segment [88 - 100] | Average Strength = 0.5841 |
| Predicted disorder segment [186 - 187] | Average Strength = 0.5429 |
| Predicted disorder segment [238 - 256] | Average Strength = 0.6208 |
| Predicted disorder segment [258 - 258] | Average Strength = 0.5028 |
| Predicted disorder segment [313 - 360] | Average Strength = 0.8331 |

**Phosphorylation sites of Mig1 and Msn2 (uniprot.org, accessed February, 2016):**

| Mig1 phosphorylation site | Disorder segment | Msn2 phosphorylation site | Disorder segment |
|---|---|---|---|

*Table 6 continued on next page*

*Table 6 continued*

**Phosphorylation sites of Mig1 and Msn2 (uniprot.org, accessed February, 2016):**

| Mig1 phosphorylation site | Disorder segment | Msn2 phosphorylation site | Disorder segment |
|---|---|---|---|
| S264 | [254 - 272] | S194 | [55 - 199] |
| S278 | - | S201 | - |
| T280 | - | S288 | - |
| S302 | [292 - 310] | S304 | - |
| S311 | [292 - 310] | S306 | - |
| S314 | - | S308 | - |
| S80 | [77 - 171] | S432 | - |
| S108 | [77 - 171] | S451 | - |
| S214 | [173 - 240] | S582 | [572 - 641] |
| S218 | [173 - 240] | S620 | [572 - 641] |
| S222 | [173 - 240] | S625 | [572 - 641]] |
| S303 | [292 - 310] | T627 | [572 - 641] |
| S310 | [292 - 310] | S629 | [572 - 641] |
| S350 | [327 - 386] | S633 | [572 - 641] |
| S367 | [327 - 386] | | |
| S370 | [327 - 386] | | |
| T371 | [327 - 386] | | |
| S377 | [327 - 386] | | |
| S379 | [327 - 386] | | |
| S381 | [327 - 386] | | |
| S400 | - | | |
| S402 | - | | |
| T455 | [423 - 504] | | |

DOI: https://doi.org/10.7554/eLife.27451.025

factor in mammalian cells, equivalent to a residence time on DNA of just 1 s (*Gebhardt et al., 2013*). Single GR molecules appear to bind as a homodimer complex on DNA, and slower Mig1 off rates may suggest higher order multivalency, consistent with Mig1 clusters.

Estimating nearest-neighbor distances between Mig1 promoter sites in the *S. cerevisiae* genome from the 3C model (*Figure 6D*) indicates 20–30% are <50 nm, small enough to enable different DNA segments to be linked though intersegment transfer by a single cluster (*Gowers and Halford, 2003*; *Schmidt et al., 2014*). This separation would also enable simultaneous binding of >1 target (*Figure 6E*). The proportion of loci separated by <50 nm is also consistent with the estimated proportion of immobile foci and with the proportion of cluster-occupied sites predicted from our structural model. Such multivalency chimes with the tetrameric binding of prokaryotic LacI leading to similar low promoter off rates (*Mahmutovic et al., 2015*).

Measuring the variation of electrostatic charge of residues for the amino acid sequences of both Mig1 and Msn2 (*Figure 6F*) we see that the regions in the vicinity of the Zn finger motifs for both proteins have a strong net positive charge compared to the rest of the molecule. If these regions project outwards from a multivalent transcription factor cluster, as per our hypothesized cluster model (*Figure 6E*), then the cluster surface could interact electrostatically with the negatively charged phosphate backbone of DNA to enable a 1D sliding diffusion of the protein along a DNA strand, such that the on rate for the protein-DNA interaction is largely sequence-independent in regards to the DNA. Particular details of this type of transcription factor binding to non-specific regions of DNA have been investigated at the level of single transcription factor molecules using computational simulations (*Rohs et al., 2010*), and suggest initial recognition is most likely via the DNA minor grooves where the phosphates are closer to each other, followed by subsequent

interactions between exposed residues on the transcription factor surface and nitrogen bases. This lack of sequence dependence for binding is consistent with observations from an earlier live cell single-molecule tracking study of the TetR repressor (*Normanno et al., 2015*). We also see experimental evidence for this in our study here, in that we find that the best fit model to account for fluorescence images of the nucleus under high glucose conditions is a combination of occupancy of clusters at the target genes (i.e. sequence specific) with random occupancy to other parts of the genome away from the target genes (i.e. sequence non-specific). Ultimate binding to the gene target once encountered could then be mediated through sequence-specific interactions via the Zn finger motif itself.

If the haploid genome of budding yeast, containing 12.1Mbp, is modeled as a flexible 'virtual' tube of length 4.1 mm ($12.1 \times 10^6 \times 0.34$ nm for each bp separation parallel to the double helix axis of DNA) with a circular cross-section, then we can calculate that the diameter of the tube required in principle to completely occupy the volume of a typical yeast nucleus (roughly a sphere of diameter ~2 μm) is 30–40 nm. This tube diameter, in the absence of local contributions from histone packing, is thus a rough estimate for the effective average separation of DNA strands in the nucleus (i.e. the 'mesh size'), which is very close to the diameter of clusters we observe. A multivalent transcription factor cluster thus may have only a relatively short distance to diffuse through the nucleoplasm if it dissociates from one DNA strand and then rebinds electrostatically to the next nearest strand, thereby facilitating intersegmental transfer. In this scheme, the association interaction between clusters and neighboring DNA strands is predominantly electrostatic and therefore largely, one might speculate, sequence-independent. However, sequence specificity may be relevant in generating higher-order packed structures of chromatin resulting in localized differences to the nearest neighbor separation of different DNA strands, which could therefore influence the rate at which a cluster transfers from one strand to another. In addition, there may also be localized effects of DNA topology that affect transcription factor binding, which in turn would be expected to have some sequence specificity (*Rohs et al., 2010*). Also, the off rates of cluster interactions with DNA may be more dependent on the specific sequence. For example, one might anticipate that the dissociation of translocating clusters would be influenced by the presence of obstacles, such as other proteins, already bound to DNA which in turn may have sequence specificity. In particular, bound RNA polymerases present during gene transcription at sequence specific sites could act as roadblocks to kick off translocating clusters from a DNA strand, to again facilitate intersegmental transfer.

Several previous experimental studies report observations consistent with intersegmental transfer relevant to our study here. For example, an investigation using single-molecule tracking indicated that transcription factor search times were increased if intersegmental transfer was specifically abrogated (*Elf et al., 2007*). These observations are consistent with other experiments that selectively enabled intersegmental transfer by altering DNA conformation (*Lomholt et al., 2009*; *van den Broek et al., 2008*). Also, they are consistent with biochemical measurements that transcription factors spend a high fraction of their time bound to DNA, as opposed to being in solution (*Elf et al., 2007*; *Esadze and Iwahara, 2014*). Furthermore, other light microscopy studies report direct experimental evidence for intersegmental transfer (*Gowers and Halford, 2003*; *Gowers et al., 2005*).

It is well-established from multiple studies that 3D diffusion of transcription factors in the nucleoplasm alone cannot account for the relatively rapid search times observed experimentally to find specific targets in the genome (*Berg et al., 1981*; *Mahmutovic et al., 2015*; *Halford and Marko, 2004*; *Gowers and Halford, 2003*). Constraining the dimensionality of diffusion to just 1D, as in the sliding of weakly bound transcription factors on DNA, speeds up this process, but is limited by encountering obstacles already bound to the DNA which potentially result in dissociation of the transcription factor and then slow 3D diffusion in the nucleoplasm. In our system, we speculate that the clusters we observe can slide on DNA in a largely sequence-independent manner but then can cross to neighboring DNA strands in a process likely to have some sequence dependence when an obstacle is encountered, and thus predominantly circumvent the requirement for slow 3D diffusion in the nucleoplasm. Minimizing the contribution from the slowest component in the search process may therefore result in an overall reduction in the amount of time required for a given transcription factor to find its gene target.

Extensive bioinformatics analysis of proteome disorder across a range of species suggests a sharp increase from prokaryotes to eukaryotes (*Xue et al., 2012*), speculatively due to the prokaryotic absence of cell compartments and regulated ubiquitination mechanisms lowering protection of

unfolded disordered structures from degradation (*Ward et al., 2004*). Our discovery in yeast may reveal a eukaryotic adaptation that stabilizes gene expression. The slow off rate we measure would result in insensitivity to high frequency stochastic noise which could otherwise result in false positive detection and an associated wasteful expression response. We also note that long turnover times may facilitate modulation between co-regulatory factors by maximizing overlap periods, as suggested previously for Mig1/Msn2 (*Lin et al., 2015*).

Our results suggest that cellular depletion forces due to crowding enable cluster formation. Crowding is known to increase oligomerization reaction rates for low association proteins but slow down fast reactions due to an associated decrease in diffusion rates, and have a more pronounced effect on higher order multimers rather than dimers (*Phillip and Schreiber, 2013*). It is technically challenging to study depletion forces in vivo, however there is growing in vitro and in silico evidence of the importance of molecular crowding in cell biology. A particularly striking effect was observed previously in the formation of clusters of the bacterial cell division protein FtsZ in the presence of two crowding proteins – hemoglobin and BSA (*Rivas et al., 2001*). Higher order decamers and multimers were observed in the presence of crowding agents and these structures are thought to account for as much as 1/3 of the in vivo FtsZ content. Similarly, two recent yeast studies of the high-osmolarity glycerol (HOG) pathway also suggest a dependence on gene expression mediated by molecular crowding (*Babazadeh et al., 2013*; *Miermont et al., 2013*).

The range of GFP labeled Mig1 cluster diameters in vivo of 15–50 nm is smaller than the 80 nm diameter of yeast nuclear pore complexes (*Ma and Yang, 2010*), not prohibitively large as to prevent intact clusters from translocating across the nuclear envelope. An earlier in vitro study using sucrose gradient centrifugation suggested a Stokes radius of 4.8 nm for the Mig1 fraction, that is diameter 9.6 nm, large for a Mig1 monomer (*Needham and Trumbly, 2006*) whose molecular weight is 55.5 kDa, for example that of monomeric bovine serum albumin (BSA) at a molecular weight of 66 kDa is closer to 3.5 nm (*Axelsson, 1978*). The authors ascribed this effect to a hypothetical elongated monomeric structure for Mig1. The equivalent Stokes radius for GFP has been measured at 2.4 nm (*Hink et al., 2000*), that is diameter 4.8 nm. Also, for our Mig1-GFP construct there are two amino acids residues in the linker region between the Mig1 and GFP sequences (i.e. additional length 0.7–0.8 nm). Thus the anticipated hydrodynamic diameter of Mig1-GFP is 15–16 nm. The mean observed ~7 mer cluster diameter from Slimfield data is ~30 nm, which, assuming a spherical packing geometry, suggests a subunit diameter for single Mig1-GFP molecules of ~$30/7^{1/3}$ ≈ 15.6 nm, consistent with that predicted from the earlier hydrodynamic expectations. Using Stokes law this estimated hydrodynamic radius indicates an effective viscosity for the cytoplasm and nucleoplasm as low as 2-3cP, compatible with earlier live cell estimates on mammalian cells using fluorescence correlation spectroscopy (FCS) (*Liang et al., 2009*).

One alternative hypothesis to that of intrinsically disordered sequences mediating Mig1 cluster formation is the existence of a hypothetical cofactor protein to Mig1. However, such a cofactor would be invisible on our Slimfield assay but would result in a larger measured hydrodynamic radius than we estimate from fluorescence imaging, which would be manifest as larger apparent viscosity values than those we observe. Coupled to observations of Msn2 forming clusters also, and the lack of any reported stable cofactor candidate to date, limits the cofactor hypothesis. Pull down assays do suggest that promoter bound Mig1 consists of a complex which includes the accessory proteins Ssn6 and Tup1 (*Treitel and Carlson, 1995*), however this would not explain the observation of Mig1 clusters outside the nucleus.

There may be other advantages in having a different strategy between *S. cerevisiae* and *E. coli* to achieve lowered transcriptional regulator off rate. A clue to these may lie in phosphorylation. We discovered that at least 50% of candidate serine or threonine phosphorylation sites in Mig1 and Msn2 lie in regions with high intrinsic disorder, which may have higher sequence-unspecific binding affinities to DNA (*Uversky et al., 2015*; *Toretsky and Wright, 2014*). Thus phosphorylation at sites within these regions may potentially disrupt binding to DNA, similar to observed changes to protein-protein affinity being coupled to protein phosphorylation state (*Nishi et al., 2013*). Previous studies indicate that dephosphorylated Mig1 binds to its targets (*Schüller, 2003*). Thus, intrinsic disorder may be required for bistability in affinity of Mig1/Msn2 to DNA.

Wide scale bioinformatics screening reveals a significant prevalence of intrinsic disorder in eukaryotic transcription factors (*Liu et al., 2006*). Our discovery is the first, to our knowledge, to make a link between predicted disorder and the ability to form higher-order clusters in transcription factors.

Thus, our results address the longstanding question of why there is so much predicted disorder in eukaryote transcription factors. Our observations that protein interactions based on weak intracellular forces and molecular crowding has direct functional relevance may stimulate new research lines in several areas of cell biology. For example, our findings may have important mechanistic implications for other aggregation processes mediated through intrinsic disorder interactions, such as those of amyloid plaques found in neurodegenerative disorders including Alzheimer's and Parkinson's diseases (*Uversky and Patel, 2015*). Increased understanding of the clustering mechanism may not only be of value in understanding such diseases, but could enable future novel synthetic biology applications to manufacture gene circuits with, for example, a range of bespoke response times.

## Materials and methods

### Strain construction and characterization

We developed Mig1 fluorescent protein strains based on strain YSH1351 (*Bendrioua et al., 2014*) using eGFP in the first instance and also mGFP/GFPmut3 designed to inhibit oligomerization (*Zacharias et al., 2002*), and photoswitchable mEos2 (*McKinney et al., 2009*). Mig1-mGFP and Mig1-mEos2 fusions were constructed by introducing into YSH1351 (BY4741 wild type) cells the *mGFP-HIS3* or *mEOs2-HIS3* PCR fragment flanked on its 5′ end with 50 bp sequence of *MIG1* 3′ end and 50 bp downstream of *MIG1* excluding the STOP codon. The *mEOs2-HIS3* and *mGFP-HIS3* fragment was amplified from mEOS-his plasmid (GeneArt, Life Technologies, Renfrew, UK) and pmGFP-S plasmid designed for this study by inserting the mGFP sequence into plasmid YDp-H. Modified strains in which the *SNF1* gene was deleted, *snf1Δ*, were prepared by compromising the gene with an auxotrophic marker by providing the *LEU2* fragment amplified from plasmid YDp-L and flanked with 50 bp of *SNF1* upstream and downstream sequence on 5′ and 3′ ends, respectively, directly into cells. Strains in which Snf1 kinase activity can be inhibited by 25 μM 1NM-PP1 (Cayman Chemical, Ann Arbor, Michigan, USA) in DMSO were prepared by introducing into cells a plasmid with an ATP analog-sensitive version of Snf1 with *I132G* mutation (*Rubenstein et al., 2008*). DMSO itself has been shown previously not to affect Mig1's behavior under different glucose conditions (*Shashkova et al., 2017*) similar to our own findings (*Figure 2—figure supplement 2*). All transformations were performed using the lithium acetate protocol (*Gietz and Schiestl, 2007*).

Cell doubling times of all strains were calculated (*Warringer et al., 2011*) (*Figure 1—figure supplement 2A*) based on $OD_{600}$ values obtained during cultivation in media supplemented with 4% or 0.2% glucose (Bioscreen analyser C). We quantified mRNA relative expression of the *MIG1* gene using qPCR against the constitutive actin gene *ACT1* in the wild type and the Mig1-mGFP strain in cells pre-grown in 4% glucose and then shifted to elevated (4%) and depleted (0.2%) extracellular glucose for 2 hr. mRNA isolation and cDNA synthesis were performed as described previously (*Geijer et al., 2013*).

For Msn2-GFP experiments we used the YSH2350 strain (*MATa msn2-GFP-HIS3 nrd1-mCherry-hphNT1 MET LYS*) in BY4741 background.

### Protein production and purification

His-tagged *mCherry*, *eGFP* and *mGFP* genes were amplified by PCR and cloned into pET vectors. An expression pRSET A plasmid containing 6xHis-Mig1-mGFP was obtained commercially (GeneArt, Life Technologies). *Escherichia coli* strain BL21(DE3) carrying the expression plasmid was grown in LB with 100 μg/ml ampicillin and 34 μg/ml chloramphenicol at 37°C to $OD_{600}$ 0.7. Protein expression was induced by adding isopropyl-β-D-thiogalactopyranoside (IPTG) at final concentration of 1 mM for 3 hr at 30°C. Cells were suspended in 50 mM $NaH_2PO_4$, 10 mM Tris, 300 mM NaCl, 2 mM EDTA, 0.2 mM PMSF, 0.1% β-mercaptoethanol, pH 8.0, and lysed by sonication or by three passages through a chilled Emulsiflex (Avestin, Mannheim, Germany). Extracts were cleared (24,000 g, 30 min) and filtered (pore diameter 0.45 μm; Millipore, Bedford). All proteins were purified using $Ni^{2+}$ affinity chromatography on a 5 ml HisTrap FF column (GE Healthcare, Chicago, Illinois, USA). Mig1-mGFP was eluted with a linear gradient 0–0.4 M imidazole in lysis buffer. Mig1-mGFP was further purified by size-exclusion chromatography (Superdex 200 Increase 10/300, GE Healthcare) and concentrated (50 kDa molecular weight cutoff VIVASPIN 20 concentrator). Purity of the sample was confirmed by

Coomassie stained SDS-PAGE gels (Simply Blue Safe Stain, Life Technologies, Carlsbad, California, United States).

## Media and growth conditions

Cells from frozen stocks were grown on plates with standard YPD media (10 g/l yeast extract, 20 g/l bacto-peptone, 20 g/l agar) supplemented with 4% glucose (w/v) at 30°C overnight. For the liquid cultures, the YPD was prepared as above but without agar, and the cells were grown at 30°C while shaking (180 rpm).

For transformants that carried a plasmid with mutated *SNF1* (p*SNF1-I132G*) or PP7-2xGFP (pDZ276), minimal YNB media with –URA amino acid supplement was applied. For the growth rate experiments cells were grown on 100 well plates in YNB with complete amino acid supplement and 4% glucose (w/v) until logarithmic phase, subcultured into fresh medium on a new 100 well plate and grown until logarithmic phase again. 10 µl of each culture was resuspended in 250 µl of fresh YNB medium with 4% or 0.2% glucose (w/v) on a new plate and cultivated in Bioscreen analyser C for 96 hr at 30°C or 22°C. OD measurements at 600 nm were taken every 10 min with prior shaking. Each strain was represented in sextuplicates.

For microscopy experiments on the BY4741 wild type and/or cells with genetically integrated fluorescent proteins, minimal YNB media (1.7 g/l yeast nitrogen base without amino acids and $(NH_4)_2SO_4$, 5 g/l $(NH_4)_2SO_4$, 0.79 g/l complete amino acid supplement as indicated by manufacturer) with appropriate glucose concentrations was used. In brief, cells were first streaked onto YPD plates, grown overnight at 30°C prior to culturing in liquid minimal YNB media with complete amino acid supplement and 4% glucose overnight, then sub-cultured into fresh YNB with 4% glucose for 4 hr with shaking at 30°C. Cultures were spun at 3,000 rpm, re-suspended into fresh YNB with (4%) or without (0%) glucose, immobilized in 1 µl spots onto an 1% agarose well perfused with YNB minimal media with an appropriate glucose concentration enclosed between a plasma-cleaned BK7 glass microscope coverslip and slide, which permitted cells to continue to grow and divide (*Reyes-Lamothe et al., 2010*; *Badrinarayanan et al., 2012*) while being observed for up to several hours if required. Images were acquired not longer than 2 hr after the last media switch.

## SDS-PAGE

50 ml cultures of YSH1703 transformed with centromeric pMig1-HA and pSNF1-I132G-TAP or pSNF1-TAP plasmids were grown until mid-log phase in yeast nitrogen base, 4% glucose, uracil and histidine deficient. Each culture was separated into two new cultures with 4% and 0.05% glucose, respectively, and incubated for 30 min. The following procedure was adapted from Bendrioua *et al* (*Bendrioua et al., 2014*). Cells were harvested by centrifugation (3,000 rpm, 50 s), suspended in 1 ml of 0.1M NaOH for 5 min and spun down. Pellets were suspended in 2 ml of 2M NaOH with 7% β-mercaptoethanol for 2 min and then 50% trichloroacetic acid was added. Samples were vortexed and spun down at 13,000 rpm. The pellets were washed in 0.5 ml of 1M Tris-HCl (pH 8.0), resuspended in 50 µl of 1x SDS sample buffer (62.5 mM Tris-HCl (pH 6.8), 3% SDS, 10% glycerol, 5% β-mercaptoethanol, and 0.004% bromophenol blue) and boiled for 5 min. The protein extracts were obtained by centrifuging at the maximal speed and collecting the supernatants. For western blotting, 50 µg of extracted proteins were resolved on a Criterion TGX 10% precast polyacrylamide gel, then transferred onto a nitrocellulose membrane (Trans-Blot Turbo Transfer Pack, Bio-Rad Laboratories, Hercules, California, USA) using Trans-Blot Turbo Transfer System (Bio-Rad). After transfer, the membrane was blocked in Odyssey Blocking buffer (LI-COR Biosciences, Lincoln, Nebraska, USA). Mig1 was detected using primary mouse anti-HA (1:2000) antibodies (Santa Cruz Biotechnology, Dallas,Texas, USA), then secondary goat anti-mouse IRDye-800CW (1:5000) antibodies (LI-COR Biosciences). The result was visualized using an infrared imager (Odyssey, LI-COR Biosciences), 800 nm channel.

## Native PAGE

A 50 ml culture of the YSH2862 strain was grown until mid-log phase in rich media with 4% glucose, then, 25 ml of the culture was transferred into fresh YPD with 4% glucose, and the rest into YPD with 0.05% glucose for 30 min. The cultures were harvested by centrifugation, suspended in 0.1 ml of solubilization buffer (100 mM Tris-HCl, pH 6.8, 0.1 mM $Na_3VO_4$, 1x protease inhibitor cocktail

(Roche, Mannheim, Germany), 0.1% Triton-X100). 400 µl of glass beads were added, and cells were broken by FastPrep, 6 m/s, 20 s. Protein extracts were obtained by adding 150 µl of solubilization buffer, centrifugation at 13,000 rpm, 5 min and collecting the supernatant. Protein quantification was performed by using Bradford with BSA standard (Bio-Rad). 250 µg of total protein extracts were run on a Criterion TGX Stain Free 10% precast polyacrylamide gel (Bio-Rad). Samples were diluted 1:1 with 2x Native Sample Buffer (Bio-Rad). Electrophoresis was performed at 4°C starting at 100V until the bromophenol blue line reached the end of the gel. The gel was transferred onto a nitrocellulose membrane (Trans-Blot Turbo Transfer Pack, Bio-Rad) using Trans-Blot Turbo Transfer System (Bio-Rad). After transfer, the membrane was blocked in Odyssey Blocking buffer (LI-COR Biosciences), analyzed by immunoblotting with mouse anti-GFP (1:500) antibodies (Roche) and visualized with goat anti-mouse IRDye-800CW (1:5,000) antibodies (LI-COR Biosciences) by using an infrared imager (Odyssey, LI-COR Biosciences), 800 nm channel. As a molecular weight reference, a Native-Mark Unstained Protein Standards (Invitrogen) were used.

## Slimfield microscopy

A dual-color bespoke laser excitation single-molecule fluorescence microscope was used (*Badrinarayanan et al., 2012*; *Wollman and Leake, 2015*; *Shashkova and Leake, 2017*) utilizing narrow epifluorescence excitation of 10 µm full width at half maximum (FWHM) in the sample plane to generate Slimfield illumination. GFP and mCherry excitation used co-aligned linearly polarized 488 nm and 561 nm wavelength 50 mW lasers (CoherentInc., OBIS lasers, Santa

Clara, California, USA) respectively which could be attenuated independently via neutral density filters followed by propagation through an achromatic λ/2 plate to rotate the plane of polarization prior to separation into two independent paths generated by splitting into orthogonal polarization components by a polarization splitting cube to enable simultaneous Slimfield illumination and a focused laser bleach illumination path for fluorescence recovery after photobleaching (FRAP) when required. The two paths were reformed into a single common path via a second polarization cube, circularized for polarization via an achromatic λ/4 plate with fast axis orientated at 45° to the polarization axes of each path and directed at ~6 W/cm$^2$ excitation intensity onto the sample mounted on an *xyz* nanostage (Mad City Labs, the Dane County, Wisconsin, USA) via a dual-pass green/red dichroic mirror centered at long-pass wavelength 560 nm and emission filters with 25 nm bandwidths centered at 525 nm and 594 nm (Chroma Technology Corp., Rockingham, Vermont, USA).

Fluorescence emissions were captured by a 1.49NA oil immersion objective lens (Nikon, Tokyo, Japan) and split into green and red detection channels using a bespoke color splitter utilizing a long-pass dichroic mirror with wavelength cut-off of 565 nm prior to imaging each channel onto separate halves of the same EMCCD camera detector (iXon DV860-BI, Andor Technology, UK) at a pixel magnification of 80 nm/pixel using 5 ms camera exposure time. We confirmed negligible measured crosstalk between GFP and mCherry signals to red and green channels respectively, using purified GFP and mCherry sampled in an in vitro surface immobilization assay (details below).

Three color microscopy was performed on the same microscope, using a 50 mW 532 nm wavelength laser (Obis) to excite mKO2, coupled into the same optics as before with the addition of a 532 nm notch rejection filter (Semrock, Rochester, New York, UK) in both channels of the imaging path. This allowed 1 mW of laser excitation at the sample. Due to the high copy number of plasmid expressed PP7-2xGFP and the 48 RNA loci, the 488 nm wavelength laser was attenuated to ~10µW. Each fluorophore was separately excited in the following order: mCherry, mKO2 and GFP to prevent crosstalk. mCherry and mKO2 both emit in the 'red' channel of the microscope, while GFP appears in the 'green' with very limited crosstalk.

## Microfluidics control of single cell imaging

To investigate time-resolved glucose concentration-dependent changes in Mig1-GFP localization in individual yeast cells, we used bespoke microfluidics and our bespoke control software *CellBild* (LabVIEW, National Instruments, Austin, Texas, United States), enabling cell-to-cell imaging in response to environmental glucose changes. *CellBild* controlled camera acquisition synchronized to flow-cell environmental switches via a syringe pump containing an alternate glucose environment. Microfluidic flow-chambers were based on an earlier 4-channel design (*Gustavsson et al., 2012*).

Prior to each experiment flow-chambers were wetted and pre-treated for 15 min with 1 mg/ml of concanavalin A (ConA) which binds to the glass surface of the plasma cleaned flow-chamber. Cells were introduced via a side channel and were left to bind ConA for 15 min to immobilize cells on the surface. Any remaining ConA and unbound cells were washed out and a steady flow of YNB with 0% glucose provided to one of the central channels by gravity feed. A syringe pump synchronized with image acquisition introduced YNB with 4% glucose in the second central channel. The pumped alternate environment reaches cells within 1–2 s at a flow rate of 10 μl/min, enabling rapid change between two different glucose concentrations.

Slimfield imaging was performed on a similar bespoke microscope setup at comparable laser excitation intensities and spectral filtering prior to imaging onto a Photometrics *Evolve Delta 512* EMCCD camera at 200 frames per second. Alternating frame laser excitation (ALEX) was used to minimize any autofluorescence contamination in the red channel introduced by the blue excitation light.

Around 1–4 cells were imaged in a single field of view for each glucose exchange. The same flow chamber was used for multiple fields of view such that each cell analyzed may have experienced up to four glucose exchange cycles.

## Foci detection, tracking and stoichiometry determination

Foci were automatically detected using software written in MATLAB (Mathworks) (*Miller et al., 2015*), lateral localization ~40 nm, enabling estimates of $D$ and stoichiometry. Our bespoke foci detection and tracking software objectively identifies candidate bright foci by a combination of pixel intensity thresholding and image transformation to yield bright pixel coordinates. The intensity centroid and characteristic intensity, defined as the sum of the pixel intensities inside a 5 pixel radius region of interest around the foci minus the local background and corrected for non-uniformity in the excitation field are determined by iterative Gaussian masking. If the signal-to-noise ratio of the foci, defined as the characteristic intensity per pixel/background standard deviation per pixel, is >0.4 it is accepted and fitted with a 2D radial Gaussian function to determine its sigma width, which our simulations indicate single-molecule sensitivity under typical in vivo imaging conditions (*Wollman and Leake, 2015*). Foci in consecutive image frames within a single point spread function (PSF) width, and not different in brightness or sigma width by more than a factor of two, are linked into the same track. The microscopic diffusion coefficient $D$ is then estimated for each accepted foci track using mean square displacement analysis, in addition to several other mobility parameters.

Cell and nuclear boundaries were segmented from GFP and mCherry fluorescence images respectively using a relative threshold pixel intensity value trained on simulated images of uniform fluorescence in idealized spherical compartments. An optimized threshold value of 0.3 times the mean compartment fluorescence intensity segmented the boundary to within 0.5 pixels.

The characteristic brightness of a single GFP molecule was determined directly from in vivo data and corroborated using in vitro immobilized protein assays (*Leake et al., 2006*). The intensity of tracked fluorescent foci in live cells was measured over time as described above (*Figure 1—figure supplement 3*). These followed an approximately exponential photobleach decay function of intensity with respect to time. Every oligomeric Mig1-GFP complex as it photobleaches to zero intensity will emit the characteristic single GFP intensity value, $I_{GFP}$, that is the brightness of a single GFP molecule, given in our case by the modal value of all foci intensities over time, and can potentially bleach in integer steps of this value at each sampling time point. This value of $I_{GFP}$ was further verified by Fourier spectral analysis of the pairwise distance distribution (*Leake et al., 2006*) of all foci intensities which yields the same value to within measurement error in our system.

All foci tracks found within 70 image frames of the start of laser illumination were included in the analysis and were corrected for photobleaching by weighting the measured foci intensity $I$ at a time $t$ following the start of laser illumination with a function $\exp(+t/t_b)$ to correct for the exponential photobleach decay $I_0\exp(-t/t_b)$, of each intensity trace with a fixed time constant, where $I_0$ is the initial unbleached intensity. This photobleach time constant $t_b$ was determined from exponential decay fits to the foci intensities and whole cell intensities over time to be 40 ± 0.6 ms. Stoichiometries were obtained by dividing the photobleach estimate for the initial intensity $I_0$ of a given foci by the characteristic single GFP molecule brightness value $I_{GFP}$.

Autofluorescence correction was applied to pool quantification by subtracting the red channel image from the green channel image multiplied by a correlation factor. By comparing wild type and

GFP cell images we confirmed that when only the GFP exciting 488 nm wavelength laser was used the green channel image contained fluorescence intensity from GFP and autofluorescence, while the red channel contains only autofluorescence pixels, consistent with expectations from transmission spectra of known autofluorescent components in yeast cells. We measured the red channel autofluorescence pixels to be linearly proportional to the green channel autofluorescence pixels. The scaling factor between channels was determined by Slimfield imaging of the wild type yeast strain (i.e. non GFP) under the same conditions and comparing intensity values pixel-by-pixel in each channel. A linear relationship between pixels was found with scaling factor of $0.9 \pm 0.1$.

Copy numbers of Mig1-GFP of the pool component were estimated using a previously developed CoPro algorithm (*Wollman and Leake, 2015*). In brief, the cytoplasmic and nuclear pools were modelled as uniform fluorescence over spherical cells and nuclei using experimentally measured radii. A model PSF was integrated over these two volumes to create model nuclear and cytoplasmic images and then their relative contributions to the camera background and autofluorescence corrected GFP intensity image determined by solving a set of linear equations for each pixel. Dividing the contributions by the characteristic single GFP molecule intensity and correcting for out-of-plane foci yields the pool concentration.

Stoichiometry distributions were rendered as objective kernel density estimations (*Leake et al., 2006*) using a Gaussian kernel with bandwidth optimized for normally distributed data using standard MATLAB routines.

## Stochastic optical reconstruction microscopy (STORM)

To photoswitch Mig1-mEos2, a 405 nm wavelength laser (Coherent Obis), attenuated to ~1 mW/cm$^2$ was used in conjunction with the 488 nm and 561 nm lasers on the Slimfield microscope, similar to previous super-resolution imaging of yeast cells (*Puchner et al., 2013*). The 405 nm laser light causes mEos2 to photoswitch from a green (excitable via the 488 nm laser) to a red (excitable by the 561 nm laser) fluorescent state. Using low intensity 405 nm light generates photoactive fluorophore foci, photobleached by the 561 nm laser at a rate which results in an approximately steady-state concentration density in each live cell studied. The bright foci were tracked using the same parameters and criteria for spot acceptance as the Slimfield data. The tracks were then used to generate a super-resolved image heat map with 20 nm pixel size by the summation of 2D Gaussian functions at each sub-pixel. Here, we assumed a sigma width of the 2D Gaussian function of 40 nm to match the measured lateral precision following automated particle tracking of Mig1-mEos2 foci (*Wollman and Leake, 2015*).

## Fluorescent protein brightness characterization

Fluorescent protein maturation time characterisations are described in more detail at Bio-Protocol (*Shashkova et al., 2018*). We used a surface-immobilization assay described previously (*Badrinarayanan et al., 2012*; *Wollman and Leake, 2015*) employing antibody conjugation to immobilize single molecules of GFP respectively onto the surface of plasma-cleaned BK7 glass microscope coverslips and imaged using the same buffer medium and imaging conditions as for live cell Slimfield experiments, resulting in integrated single-molecule peak intensity values for mGFP of $4,600 \pm 3000$ (±half width half maximum, HWHM) counts. Similar experiments on unmodified purified Clontech eGFP generated peak intensity values of $4,700 \pm 2000$ counts, statistically identical to that of mGFP (Student *t*-test, p=0.62) with no significant indication of multimerization effects from the measured distribution of foci intensity values. Similarly, Slimfield imaging and foci stoichiometry analysis on Mig1-mGFP and Mig1-eGFP cell strains were compared in vivo under high and low glucose conditions in two separate cell strains, resulting in distributions which were statistically identical (Pearson's $\chi^2$ test comparing KDEs, *Figure 1—figure supplement 2E and F*). These results indicated no measurable differences between multimerization state or single-molecule foci intensity between mGFP and eGFP which enabled direct comparison between Mig1-eGFP cell strain data obtained from preliminary experiments here and from previous studies (*Bendrioua et al., 2014*).

Maturation effects of mCherry and GFP were investigated by adding mRNA translation inhibitor antibiotic cycloheximide, final concentration 100 μg/ml, for 1 hr (*Hartwell et al., 1970*), photobleaching cells, then monitoring any recovery in fluorescence as a metric for newly matured fluorescent material in the cell. Cells were prepared for microscopy as before but using cycloheximide in all

subsequent preparation and imaging media and imaged using a commercial mercury-arc excitation fluorescence microscope Zeiss Axiovert 200M (Carl Zeiss MicroImaging GmbH, Jena, Germany) onto an ApoTome camera using a lower excitation intensity than for Slimfield imaging but a larger field of view, enabling a greater number of cells to be imaged simultaneously.

Surface-immobilized cells using strain YSH2863 were photobleached by continuous illumination for between 3 min 40 s to 4 min until dark using separate filter sets 38HE and 43HE for GFP and mCherry excitation, respectively. Fluorescence images were acquired at subsequent time intervals up to 120 min and analyzed using AxioVision software (*Figure 1—figure supplement 2C*). The background-corrected total cellular fluorescence intensity was quantified at each time point for each cell using ImageJ software. Comparison between Mig1-GFP fluorescence signal and the green channel signal from the parental strain BY4741, and the Nrd1-mCherry signal and the red channel signal from the parental strain, indicate fluorescence recovery after correction above the level of any auto-fluorescence contributions of <15% for GFP and mCherry over the timescale of our experiments, consistent with previous estimates of in vivo maturation times for GFP and mCherry (*Badrinarayanan et al., 2012*; *Leake et al., 2006*; *Khmelinskii et al., 2012*).

## Characterizing Mig1-GFP clusters in vitro

Using Slimfield microscopy under the same imaging conditions as for live cell microscopy we measured the fluorescent foci intensity of 1 µg/ml solutions of purified Mig1-mGFP and mGFP using the normal imaging buffer of PBS, compared with the imaging buffer supplemented with 1 kDa molecular weight PEG at a concentration of 10% (w/v) used to reproduce cellular depletion forces (*Phillip and Schreiber, 2013*; *Warringer et al., 2011*).

## Circular dichroism measurements

Purified Mig1-mGFP was placed in 25 mM $Na_2HPO_4$, pH 7.0, by buffer exchange procedure with a Pur-A-Lyser Maxi dialysis Kit (Sigma Aldrich, St. Louis, Missouri, United States) for 3 hr at 4°C with constant stirring in 500 ml buffer. Circular dichroism measurements were performed on a Jasco J810 circular dichromator with Peltier temperature control and Biologic SFM300 stop-flow accessory on 0.16 mg/ml Mig1-mGFP samples with or without 20% PEG-1000 at 20°C, from 260 to 200 nm, a 2 nm band width, 2 s response time, at the speed of 100 nm/min. The resulting spectrum represents the average of 5 scans, indicating a typical SD error of ~0.1 mdeg ellipticity. Spectra from 25 mM $Na_2HPO_4$ and 25 mM $Na_2HPO_4$ with 20% (w/v) PEG were used as a background and subtracted from spectra of Mig1-mGFP without or with 20% (w/v) PEG respectively.

## Immuno-gold electron microscopy

Cells for Mig1-GFP and Msn2-GFP strains as well as the wild type control strain containing no GFP were grown using the same conditions as for Slimfield imaging but pelleted down at the end of growth and prepared for immuno electron microscopy using an adaptation of the Tokuyasu cryosectioning method (*Tokuyasu, 1973*) following the same protocol that had been previously optimized for budding yeast cells (*Griffith et al., 2008*) to generate ~90 nm thick cryosections, with the exception that the sections were picked up on a drop of 2.3M sucrose, placed on the grid, then floated down on PBS, and then immunolabeled immediately, rather than storing on gelatine as occurred in the earlier protocol. The grids used were nickel, with a formvar/carbon support film. In brief, the immunolabeling protocol used a 0.05M glycine in PBS wash of each section for 5 min followed by a block of 10% goat serum in PBS (GS/PBS) pre-filtered through a 0.2 µm diameter filter. Then an incubation of 1 hr with the primary antibody of rabbit polyclonal anti-GFP (ab6556, Abcam, Cambridge, UK) at 1 in 250 dilution from stock in GS/PBS. Then five 3 min washes in GS/PBS. Then incubation for 45 min with the goat anti-IgG-rabbit secondary antibody labeled with 10 nm diameter gold (EM.GAR10, BBI solutions) at a dilution of 1 in 10 from stock. Sections were then washed five more times in GS/PBS prior to chemical fixation in 1% glutaraldehyde in sodium phosphate buffer for 10 min, then washed in $dH_2O$ five times for 3 min each and negative-stained using methyl cellulose 2% in 0.4% uranyl acetate, and then washed twice more in $dH_2O$ prior to drying for 10 min. Drop sizes for staining, blocking and washing onto sections were 50 µl, while antibody incubations used 25 µl drops, all steps performed at room temperatures.

Electron microscopy was performed on these dried sections using a 120kV Tecnai 12 BioTWIN (FEI) electron microscope in transmission mode, and imaged onto an SIS Megaview III camera. From a total of ~150 control cells containing no GFP we could detect no obvious signs of gold labeling. Using approximately the same number of cells for each of the Mig1-GFP and Msn2-GFP strains all images showed evidence for at least one gold foci labeling in the cytoplasm, though labeling was largely absent from the nucleus possibly due to poor antibody accessibility into regions of tightly packed DNA since the combined Stokes radii from the primary and secondary antibodies is comparable to the mean effective DNA mesh size in the yeast nucleus of a few tens of nm (see Discussion section). We estimate that the thin cryosections occupy ~2.5% of the volume of an average yeast cell and so based on our copy number estimates from fluorescence microscopy in the accessible cytoplasmic compartment the maximum number of GFP available for labelling in each cryosection is ~20 molecules. We observed a range of 1–8 gold foci in total per cell across the GFP datasets and so the overall labelling efficiency in these experiments is low at typically 20% or less. However, we observed 10 cells from a set of ~150 from each of the Mig-GFP and Msn2-GFP strains (i.e. ~7% of the total) which showed >1 gold foci clustering together inside an area of effective diameter ~50 nm or less, with up to seven gold foci per cluster being observed.

## Bioinformatics analysis and structural modeling

Bioinformatics analysis was used to identity candidate promoter sequences in the budding yeast genome. The Mig1 target pattern sequence was identified based on 14 promoter sequences (*Lundin et al., 1994*) using the IUPAC nucleotide code. The entire *S. cerevisiae* S288c genome was scanned in order to find all sequences that matched the pattern. The scanning was performed by RNABOB software (*Riccitelli and Lupták, 2010*), and collated for any further analysis and identification of the sequences lying within promoter regions. All information regarding *S. cerevisiae* genes was obtained from SGD database (http://yeastgenome.org/).(*Cherry et al., 2012*)

Using a consensus structural model for the budding yeast chromosome based on 3C data (*Duan et al., 2010*) we explored various different models of Mig1 binding to the putative promoter sequence identified from the bioinformatics analysis. We generated simulated images from these models adding experimentally realistic levels of signal and noise, and ran these data through the same foci detection and analysis software as for the real live cell data using identical parameters throughout. We then compared these results to the measured experimental stoichiometry (*Figure 4C*). Monomer models assume that a single Mig1 molecule binds to a target promoter site, whereas cluster models assume that a cluster comprising 7 Mig1 molecules (based on our observations of stoichiometry periodicity) binds a single target promoter. Copy number analysis indicated 190 Mig1 molecules per cell on average associated with foci. In the monomer model (*Figure 4C*) all 109 promoter sites were assigned a Mig1 molecule and the remaining 81 Mig1 molecules were placed randomly in the 222 remaining Mig1 target binding sites within the rest of the genome. In the DNA cluster model (*Figure 4—figure supplement 1*) we randomly assigned the observed 190 Mig1 molecules in foci into just 27 clusters to Mig1 target promoter sites. We also tested two nuclear envelope (NE) variants of both models, to account for the trans-nuclear tracks: here, typically ~7 Mig1 were observed translocating from the nucleus to the cytoplasm at *glucose* (+) within the microscope's depth of field; extrapolating this value over the whole nucleus this indicates ~130 Mig1 molecules within the nucleus but less than a single PSF width from the nuclear envelope prior to export to the cytoplasm. We simulated this effect using either 130 Mig1 molecules as Mig1 monomers or as 18 (i.e. ~130/7) 7-mer clusters at random 3D coordinates at the simulated nuclear envelope position in the 3C model. Finally, to generate the best fit Mig1 cluster model, we obtained an optimized fit to the data with a mixed population model with 75% of cells in the NE cluster model and 25% in the DNA cluster model. We note here that the monomer model can produce higher apparent stoichiometry due to the increased density of resulting foci (although the same density of Mig1).

We used bioinformatics to investigate the extent of intrinsic disorder in the amino acid sequence of budding yeast Mig1 and Msn2 proteins as well as the *E. coli lac* repressor LacI, employing the Predictor of Natural Disordered Regions (PONDR) algorithm (*Obradovic et al., 2005*) (online tool http://www.pondr.com/cgi-bin/PONDR/pondr.cgi) with a VL-XT algorithm. We also used the secondary structure prediction algorithm of PyMOL (http://www.pymolwiki.org/index.php/Dss) to highlight disordered and structured regions and display the unfolded protein chain, and used PSI-BLAST

multiple sequence alignment to determine conserved structural features of Mig1 for the Zn finger motif in combination with the DISOPRED (*Ward et al., 2004*) algorithm as a comparison to PONDR, which produced very similar results (online tool http://www.yeastrc.org/pdr/).

## Oligomerization state of Mig1-GFP in the 'pool'

Experimental in vitro assays of surface immobilized GFP coupled to simulations trained on these single-molecule intensity measurements but using noise levels comparable to in vivo cellular imaging conditions (*Wollman and Leake, 2015*) indicate single-molecule sensitivity of GFP detection under our millisecond imaging conditions. However, if the nearest neighbor separation of individual GFP 'foci' are less than the optical resolution limit $w$ of our microscope (which we measure as ~230 nm for GFP imaging) then distinct fluorescent foci will not be detected and instead will be manifest as a diffusive 'pool'.

If each GFP 'foci' in the pool has a mean stoichiometry $S$ then the mean number of GFP foci, $F$, in the pool is $n_{pool}/S$ and the 'pool' condition for nearest neighbor foci separation $s$ indicates that $s < w$.

The estimated range of mean total pool copy number from nucleus and cytoplasm combined, $n_{pool}$, is ~590–1,100 molecules depending on extracellular glucose conditions. Approximating the cell volume as equal to the combined volumes of all uniformly separated foci in the pool (equal to the total number of foci multiplied by the volume of an equivalent sphere of radius $r$) indicates that $F.4\pi r^3/3 = 4\pi d^3/3$, thus $r = d/F^{1/3}$, where we use the mean measured cell diameter $d$ of ~5 μm.

However, mobile foci with a microscopic diffusion coefficient $D$ will diffuse a mean two-dimensional distance $b$ in focal plane of $(4D.\Delta t)^{1/2}$ in a camera sampling time window $\Delta t$ of 5 ms. Using $D \sim 6$ μm$^2$ s$^{-1}$ as a lower limit based on the measured diffusion of low stoichiometry cytoplasmic Mig1-GFP foci detected indicates $b \sim 340$ nm so the movement-corrected estimate for $s$ is $r$-$b$, thus $s < w$ indicates that $r < b + $ w, or $d/F^{1/3} < b+w$.

Therefore, $d(S/n_{pool})^{1/3} < b+w$, and $S < n_{pool}((b + w)/d)^3$. Using ~590–1,100 molecules from the measured mean range of $n_{pool}$ indicates that the upper limit for $S$ is in the range 0.8–1.4; in other words, Mig1-GFP foci in the pool are consistent with being a monomer.

## Analysis of the mobility of foci

For each accepted foci track the mean square displacement (MSD) was calculated from the optimized intensity centroid at time $t$ of $(x(t), y(t))$ assuming a tracks of $N$ consecutive image frames at a time interval $\tau = n\Delta t$ is (*Gross and Webb, 1988*; *Michalet, 2010*) where $n$ is a positive integer is:

$$MSD(\tau) = MSD(n\Delta t) = \frac{1}{N-1-n} \sum_{i=1}^{N-1-n} \left\{ [x(i\Delta t + n\Delta t) - x(i\Delta t)]^2 + [y(i\Delta t + n\Delta t) - y(i\Delta t)]^2 \right\}$$
$$= 4D\tau + 4\sigma^2$$

Here $\sigma$ is the lateral ($xy$) localization precision which we estimate as ~40 nm (*Wollman and Leake, 2015*). The microscopic diffusion coefficient $D$ was then estimated from the gradient of a linear fit to the first four time interval data points of the MSD vs $\tau$ relation for each accepted foci track.

To determine the proportion of mobile and immobile Mig1-GFP fluorescent foci we adapted an approach based on cumulative probability-distance distribution analysis (*Gebhardt et al., 2013*). Here we generated cumulative distribution functions (CDFs) for all nuclear and cytoplasmic tracks, such that the CDF in each dataset is the probability distribution function $p_c$ associated with $r^2$, the square of the displacement between the first and second data points in each single track, which was generated for each track by calculating the proportion of all tracks in a dataset which have a value of $r^2$ less than that measured for that one track. The simplest CDF model assumes a Brownian diffusion propagator function $f(r^2)$ for a single effective diffusion coefficient component of:

$$f(r^2) = \frac{1}{4\pi D\Delta t} exp\left(\frac{r^2}{4D\Delta t}\right)$$

Here, $D$ is the effective diffusion coefficient and $\Delta t$ is image sampling time per frame (i.e. in our case 5 ms). This gives a CDF single component solution of the form:

$$p_c(r^2) = 1 - exp\left(\frac{r^2}{4D\Delta t}\right)$$

We investigated both single and more complex multi-component CDF models using either 1,2 or 3 different $D$ values in a weighted sum model of:

$$p_c(r^2) = \sum_{i=1}^{n} A_i \left(1 - exp\left(\frac{r^2}{4D_i\Delta t}\right)\right)$$

Here $n$ is 1, 2 or 3. Multi-component fits were only chosen if they lowered the reduced $\chi^2$ by >10%. For cytoplasmic foci at *glucose* (+/-) and for nuclear foci at *glucose* (−) this indicated single component fits for diffusion coefficient with a $D$ of ~1–2 μm$^2$/s, whereas nuclear foci at *glucose* (+) were fitted using two components of $D$, ~20% with a relatively immobile component, $D \sim$ 0.1–0.2 μm$^2$/s, and the remainder a relatively mobile component, $D \sim$ 1–2 μm$^2$/s, while using three components produced no statistically significant improvement to the fits. These values of $D$ agreed to within experimental error to those obtained using a different method which fitted two analytical Gamma functions to the distribution of all calculated microscopic diffusion coefficients of tracked foci in the nucleus at *glucose* (+), which assumed a total probability distribution function $p_\gamma$ of the form: (*Stracy et al., 2015*)

$$p_\gamma(x, D) = \sum_{i=1}^{2} \frac{A_i(m/D)^m x^{n-1} exp(-mx/D)}{(m-1)!}$$

Here, $m$ is the number of steps in the MSD *vs* $\tau$ trace for each foci track used to calculate $D$ (i.e. in our instance $m = 4$).

We also probed longer time scale effects on foci mobility for each accepted foci trajectory. Here, average MSD values were generated by calculating mean MSD values for corresponding time interval values across all foci trajectories in each dataset, but pooling traces into low stoichiometry ($\leq$20 Mig1-GFP molecules per foci) and high stoichiometry (>20 Mig1-GFP molecules per foci). We compared different diffusion models over a 30 ms time interval scale, corresponding to the shortest time interval range from any of the mean MSD trace datasets.

We found in all cases that mean MSD traces could be fitted well ($\chi^2$ values in the range 1–12) using a subdiffusion model of precision-corrected MSD = $4\sigma^2 + 4K\tau^\alpha$, where $\alpha$ the anomalous diffusion parameter and $K$ is the transport parameter, analogous to the diffusion coefficient $D$ in pure Brownian diffusion. Optimized fits indicated values of $K$ in the range 0.08–0.2 μm$^2$/s and those for $\alpha$ of ~0.4–0.8. Corresponding fits to a purely Brownian diffusion model (i.e. $\alpha = 1$) generated much poorer fits ($\chi^2$ values in the range 4–90).

We used both short timescale CDF analysis and longer timescale MSD analysis of Mig1 tracks to try to gain as complete a picture of Mig1 mobility as possible. Short timescales avoid bias from photobleaching and diffusion out of the focal plane but longer timescales sample more of the cellular environment.

### Analyzing trans-nuclear tracks

The segmentation boundary output for the nucleus was fitted with a smoothing spline function, with smoothing parameter p=0.9992 to sub-pixel precision. Trajectories which contained points on either side of the nuclear boundary were considered trans-nuclear. The crossing point on the nuclear boundary was found by linearly interpolating between the first pair of points either side of the nuclear boundary. Coordinates were normalized to this point and the crossing time and were rotated such that $y'$ and $x'$ lie perpendicular and parallel to the membrane crossing point.

### Investigating Mig1-GFP molecular turnover

Turnover of Mig1-GFP was investigated using fluorescence recovery after photobleaching (FRAP). In brief a 200 ms 10 mW focused laser beam pulse of lateral width ~1 μm was used to photobleach the fluorescently-labelled nuclear contents on a cell-by-cell basis and then $\leq$10 Slimfield images were recorded over different timescales spanning a range from 100 ms to ~1,000 s. The copy number of pool and foci in each image at subsequent time points $\underline{t}$ post focused laser bleach was determined

as described and corrected for photobleaching. These post-bleach photoactive Mig1-GFP copy number values $C(t)$ could then be fitted using a single exponential recovery function:

$$C(t) = C(0)(1 - exp(-t/t_R))$$

Where $t_R$ is the characteristic recovery (i.e. turnover) time (*Reyes-Lamothe et al., 2010*). These indicated a value of 133 ± 20 s (±SEM) for nuclear foci at glucose (+), and 3 ± 14 s for nuclear pool at *glucose* (+) and (−).

## Modeling the effective diameter of clusters

The effective diameter $d$ of a cluster was estimated from the measured point spread function width $pf_{foci}$ (defined at twice sigma value of the equivalent Gaussian fit from our single particle tracking algorithm) corrected for the blur due to particle diffusion in the camera exposure time of $\Delta t$ as:

$$d = p_{foci} - p_{GFP} - \sqrt{4D\Delta t}$$

Where $D$ is the measured microscopic diffusion coefficient for that track and $p_{GFP}$ is the measured point spread function width of surface-immobilized GFP (i.e. twice the sigma width of 230 nm measured in our microscope, or 460 nm). We explored a heuristic packing model of $d \sim S^a$ for Mig1-GFP monomers in each cluster, such that a tightly packed spherical cluster of volume $V$ composed of $S$ smaller ca. spherical monomers each of volume $V_1$ and diameter $d_1$ varied as $V = S \cdot V_1$ thus $4\pi(d/2)^3 = S.4\pi(d_1/2)^3$, thus in the specific instance of a perfect spherical cluster model $a = 1/3$.

In principle, for general shapes of clusters for different packing conformations we expect $0 \leq a \leq 1$ such that for example if clusters pack as a long, thin rod of Mig1 monomers which rotates isotropically during time $\Delta t$, then $a = 1$. Whereas, if Mig1 monomers bind to a putative additional 'anchor' type structure to occupy available binding sites in forming a cluster, such that the size of the cluster does not significantly change with $S$ but is dependent on the size of the putative anchor structure itself, then $a = 0$. Our optimized fits indicate $a = 0.32 \pm 0.06$ (±SEM), that is consistent with an approximate spherical shape cluster model.

## Modeling the probability of overlap in in vitro fluorescent protein characterization

The probability that two or more fluorescent protein foci are within the diffraction limit of our microscope in the in vitro characterization assays was determined using a previously reported Poisson model (*Llorente-Garcia et al., 2014*) to be ~10% at the in vitro protein concentrations used here. Such overlapping fluorescent proteins are detected as higher apparent stoichiometry foci.

## PP7 RNA labelling and overlap integral

Similar Slimfield microfluidics experiments were performed on Mig1-mCherry and Mig1-mCherry ΔZnf strains containing 24 transcriptional reporter PP7 markers on the GYS1 gene and transformed with plasmids for the PP7 protein labelled with 2 GFPs. Mig1 foci are present at *glucose* (+) and upon switching to *glucose* (−) PP7 foci appear in similar locations to the Mig1 foci. Although Mig1 foci are mobile, the microscopic diffusion coefficient $D$ for immobilized Mig1 is a putative overestimate for the equivalent $D$ of the underlying target gene loci, 0.15 μm$^2$/s from CDF. The plateau of the MSD *vs* tau plot in *Figure 3B* gives an estimate of the gene loci mobility range in space (although still an overestimate) and is ~0.05 μm$^2$. The square root of this is less than PSF width, and so colocalization between Mig1 and PP7 foci is expected.

The extent of colocalization between Mig1-mCherry and PP7-GFP detected foci was determined by calculating the overlap integral between each pair, whose centroids were within 5 pixels of each other. Assuming two normalized, 2D Gaussian intensity distributions g1 and g2, for green and red foci respectively, centered around $(x_1, y_1)$ with sigma width $\sigma_1$, and around $(x_2, y_2)$ with width $\sigma_2$, the overlap integral $v$ is analytically determined as:

$$v = \exp\left(-\frac{\Delta r^2}{2(\sigma_1^2 + \sigma_2^2)}\right)$$

Where

$$\Delta r^2 = \left(x_1^2 - x_2^2\right)^2 + \left(y_1^2 - y_2^2\right)^2$$

Previous studies have used an overlap integral of over 0.75 as a criteria for colocalization (*Llorente-Garcia et al., 2014*).

## Software and DNA sequence access

All our bespoke software developed, and Mig1 secondary structure prediction 3D coordinates pymolMig1.pdb, are freely and openly accessible via https://sourceforge.net/projects/york-biophysics/ (*Wollman, 2016*) (copy archived at https://github.com/elifesciences-publications/york-biophysics). The bespoke plasmid sequence information for the GFP reporter is openly accessible via https://www.addgene.org/75360/.

## Statistical tests and replicates

All statistical tests used are two-sided unless stated otherwise. For Slimfield imaging each cell can be defined as a biological replicate sampled from the cell population. We chose sample sizes of at least 30 cells which generated reasonable estimates for the sampled stoichiometry distributions, similar to those of previous in vivo Slimfied studies (*Reyes-Lamothe et al., 2010*). Technical replicates are not possible with the irreversible photobleaching assay, however, the noise in all light microscopy experiments has been independently characterized for the imaging system used previously (*Wollman and Leake, 2015*).

## Acknowledgements

Supported by the Biological Physical Sciences Institute, Royal Society, MRC (grant MR/K01580X/1), BBSRC (grant BB/N006453/1), Swedish Research Council and European Commission via Marie Curie-Network for Initial training ISOLATE (Grant agreement nr: 289995) and the Marie Curie Alumni Association. We thank Magnus Alm Rosenblad and Sarah Shammas for assistance with RNABOB and PONDR, Marija Cvijovic and Michael Law for help with qPCR data analysis, Andrew Leech and Meg Stark for help with CD and EM. Thanks to Mark Johnston (CU Denver) for donation of Mig1 phosphorylation mutant plasmid, and Michael Elowitz (Caltech) for donation of the Mig1/Msn2/PP7 and Zn finger deletion strain.

## Additional information

### Funding

| Funder | Grant reference number | Author |
|---|---|---|
| Biotechnology and Biological Sciences Research Council | grant BB/N006453/1 | Adam JM Wollman<br>Mark C Leake |
| Biological Physical Sciences Institute | 50093701 | Adam JM Wollman<br>Sviatlana Shashkova<br>Erik G Hedlund<br>Mark C Leake |
| European Commission | 289995 | Sviatlana Shashkova<br>Erik G Hedlund |
| Royal Society | Newton International Fellowship (NF160208) | Sviatlana Shashkova |
| Swedish Research Council | University of Gothenburg research infrastructure funding | Sviatlana Shashkova<br>Rosmarie Friemann<br>Stefan Hohmann |
| Medical Research Council | MR/K01580X/1 | Mark C Leake |

The funders had no role in study design, data collection and interpretation, or the decision to submit the work for publication.

## Author contributions
Adam JM Wollman, Resources, Software, Formal analysis, Investigation, Visualization, Methodology, Writing—original draft, Writing—review and editing; Sviatlana Shashkova, Resources, Formal analysis, Investigation, Visualization, Methodology, Writing—original draft, Writing—review and editing; Erik G Hedlund, Formal analysis, Investigation, Visualization, Methodology, Writing—original draft, Writing—review and editing; Rosmarie Friemann, Investigation, Methodology, Writing—original draft, Writing—review and editing; Stefan Hohmann, Conceptualization, Supervision, Funding acquisition, Writing—original draft, Writing—review and editing; Mark C Leake, Conceptualization, Data curation, Supervision, Funding acquisition, Visualization, Writing—original draft, Project administration, Writing—review and editing

## Author ORCIDs
Adam JM Wollman http://orcid.org/0000-0002-5501-8131
Sviatlana Shashkova https://orcid.org/0000-0002-4641-3295
Mark C Leake http://orcid.org/0000-0002-1715-1249

## Decision letter and Author response
Decision letter https://doi.org/10.7554/eLife.27451.029
Author response https://doi.org/10.7554/eLife.27451.030

# Additional files

## Supplementary files
• Transparent reporting form
DOI: https://doi.org/10.7554/eLife.27451.026

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
