## [Decision Letter]

Thank you for submitting your article "Transcription factor clusters regulate genes in eukaryotic cells" for consideration by *eLife*. Your article has been reviewed by three peer reviewers, one of whom is a member of our Board of Reviewing Editors, and the evaluation has been overseen by Jessica Tyler as the Senior Editor. The following individual involved in review of your submission has agreed to reveal his identity: Serge Pelet (Reviewer #3).

The reviewers have discussed the reviews with one another and the Reviewing Editor has drafted this decision to help you prepare a revised submission.

Summary:

In this article, the authors present a detailed analysis of the dynamics of MIG1, a transcription factor that plays a key role in glucose sensing in yeast. Using a combination of live-cell single-molecule tracking, bioinformatics analysis, and in vitro assays, they unveil a MIG1 clustering mechanisms involved in the response to glucose. In recent years, partly thanks to single-molecule tools, the role of transcription-factor clustering in transcriptional regulation has become an important topic of study. The present work adds to this growing list and in particular identifies a clustering mechanism through the depletion forces and via disordered domains of the protein. The manuscript is clearly written and the results will be of interest to the community. However, before publication is considered, a number of concerns will need to be addressed in relation to quantification, statistics, and controls.

Essential revisions:

It is not clear how the stoichiometry of molecules within a single focus was determined. The authors mention the measurement of single GFP photobleaching (Results paragraph three), but don't show any data. The manuscript will have to include raw trajectories and their analysis, showing the intensity distributions for single GFP bleaching steps measured intracellularly. Without these data, none of the conclusions are supported.

The authors confirm the formation of clusters by performing in vitro experiments under conditions that mimic intracellular crowding. The same focus tracking analysis is used as in the in vivo experiments. However, no raw imaging data is provided to visually confirm focus formation

The quantified data of nuclear and cytoplasmic enrichment presented in Figure 1 don't seem to match the image of the single cell. According to the image (and data from Bendrioua et al., 2014), Mig1 accumulates and remains in the nucleus for at least 400s. While the quantification reveals a drop in Mig1 nuclear level and an enrichment in Mig1 cytoplasmic level after 200 ms. Also after the switch to glucose (-) no sharp drop in nuclear level is observed in the quantified curve as expected from the image and previous quantifications (Bendrioua et al., 2014).

According to the data in Table 2, there is an increase in total fluorescence level going from the glucose (+) to the glucose (-) conditions for Mig1 and Msn2. Can these changes be due to an expression of the protein? How long after the switch to the low C-source medium are these values quantified? In addition, it has been reported that GFP is sensitive to pH and pH changes upon glucose starvation (Roberts et al., Sci. Rep. 2016). Have the authors experienced any difference in GFP brightness between conditions?

The Zn finger mutation of Mig1 seems an interesting control for many of the experiments performed. This strain is however not included in the strain list. Since the author claim that nuclear accumulation is governed by retention in the nucleus due to the binding of DNA. The ZF∆ mutant would be a good control. The overlap between Mig1 dots and PP7 foci should also disappear in this mutant.

In Figure 3, the authors present data on the constrained diffusion of the foci due to nuclear anchoring of Mig1. If I understood properly, the deviation form the dashed line represents the constrained diffusion. However, I don't see a noticeable difference between the deviation observed for small or large foci or even cytoplasmic ones. Therefore I don't understand how this can be interpreted as a effect of the DNA anchoring of the Mig1 clusters. Again a ZF∆ Mig1 would be an interesting control to measure.

The authors present data on the in vitro oligomerisation of Mig1-GFP triggered by PEG addition. As a control, they use mGFP alone. First of all, this control should be performed with the same fluorescent protein for the TF bound and the GFP alone experiment. In addition, the number of measured fluorescent dots has to be the same in both cases. However 1000 dots where measured for the Mig1-GFP versus 100 for the GFP. Since the high stoichiometry clusters are found with <1% probability, the chance of observing a single high stoichiometry cluster is lower than 1 for the control experiment.

I'm not a specialist of electron microscopy, however I find a few elements in this experiment puzzling. The authors state that they measured 150 cells and only in 10 of them did they see a cluster of Mig1 or Msn2. How many untagged cells did they measure to make sure that this phenomena is not present in untagged strains? Why don't we see individual Mig1 or Msn2 molecules labels with the gold particles in these images. Also no data of Mig1 nuclear cluster has been obtained. These low statistics makes me wonder about the quality of this dataset.

---

## [Author Response]

Essential revisions:It is not clear how the stoichiometry of molecules within a single focus was determined. The authors mention the measurement of single GFP photobleaching (Results paragraph three), but don't show any data. The manuscript will have to include raw trajectories and their analysis, showing the intensity distributions for single GFP bleaching steps measured intracellularly. Without these data, none of the conclusions are supported.

We have added a new Figure Supplement to Figure 1, Figure 1—figure supplement 3 showing raw intensity vs time traces for long lived individual nuclear foci (A), cytoplasmic clusters (B) and single in vivo Mig1-GFP photobleach steps (C). The step like nature of these data is further highlighted using an edge preserving Chung-Kennedy^1^ filter. The intensity distribution of single in vivoMig1-GFP is shown in (D). The mode of this distribution gives *I_GFP_*used to calculate the stoichiometry of individual Mig1-GFP foci by dividing the initial photobleached corrected foci intensity by *I_GFP,_*as outlined in Materials and methods, Foci detection, tracking and stoichiometry determination. Figure 1—figure supplement 3 is now referred to in the text.

The authors confirm the formation of clusters by performing in vitro experiments under conditions that mimic intracellular crowding. The same focus tracking analysis is used as in the in vivo experiments. However, no raw imaging data is provided to visually confirm focus formation

We now include the raw image data. Figure 4—figure supplement 1 has been split into 3 parts. Slimfield fluorescence micrographs for in vitroMig1-GFP and Mig1-GFP PEG induced clusters have been added to Figure 4—figure supplement 2.

The quantified data of nuclear and cytoplasmic enrichment presented in Figure 1 don't seem to match the image of the single cell. According to the image (and data from Bendrioua et al., 2014), Mig1 accumulates and remains in the nucleus for at least 400s. While the quantification reveals a drop in Mig1 nuclear level and an enrichment in Mig1 cytoplasmic level after 200 ms. Also after the switch to glucose (-) no sharp drop in nuclear level is observed in the quantified curve as expected from the image and previous quantifications (Bendrioua et al., 2014).

The reviewer is correct that the image data presented in Figure 1 appeared not to match the copy number trace from the same panel. This figure has been amended to include 2 extra sets of Slimfield microscopy images highlighting the relatively high degree of inter-cellular variation in these data. This has also been highlighted in the text. Our data differ from the experiments of Bendrioua et al^2^, specifically their Figure 3, in two ways. Firstly, they monitor the nuclear/cytoplasmic pixel intensity ratio without deconvolution such that their intensity measurements in each compartment are not separated and so do not directly correlate with our actual copy number measures. Secondly, they monitored many cells at once such that each cell had experienced only one glucose cycle. For our high magnification, high precision work here our microscope was only capable of imaging up to 4 cells simultaneously on the same camera pixel array, so cells included in the copy number analysis experienced up to 4 glucose exchange cycles. This changes the nature and level of response consistent with Bendrioua et al. Figure 2 which shows an exponential decrease in response with a second cycle (and others^3^). The Materials and methods section, “Microfluidics control of single cell imaging”, has been amended to reflect this

According to the data in Table 2, there is an increase in total fluorescence level going from the glucose (+) to the glucose (-) conditions for Mig1 and Msn2. Can these changes be due to an expression of the protein? How long after the switch to the low C-source medium are these values quantified? In addition, it has been reported that GFP is sensitive to pH and pH changes upon glucose starvation (Roberts et al., Sci. Rep. 2016). Have the authors experienced any difference in GFP brightness between conditions?

We believe that the total fluorescence level increase from *glucose* (+) to (-) is caused by differences in expression levels. We observed this in a previous study^4^ and is also suggested by qPCR measurements in Figure 1—figure supplement 2. The text has been amended in Results paragraph three to reflect this. Imaging was performed within 2 hours of switching the glucose condition and this detail added to the Materials and methods subsection “Media and growth conditions”. We did not observe any differences in fluorescent protein brightness upon glucose starvation. Figure 1—figure supplement 3 shows the in vivo Mig1-GFP intensity is the same within error for *glucose* (+)/(-). If these had been different, our method would have been insensitive, as different *I_GFP_*values would reveal any actual copy number differences between cells.

The Zn finger mutation of Mig1 seems an interesting control for many of the experiments performed. This strain is however not included in the strain list. Since the author claim that nuclear accumulation is governed by retention in the nucleus due to the binding of DNA. The ZF∆ mutant would be a good control. The overlap between Mig1 dots and PP7 foci should also disappear in this mutant.

The Zn finger deletion Mig1-mCherry strain has been added to the strain table, Table 1 and text amended in subsection “Cytoplasmic Mig1 is mobile but nuclear Mig1 has mobile and immobile states”. Figure 1—figure supplement 1 has been amended with Mig1-mCherry ΔZnf micrographs, showing wild type Mig1 localisation in both glucose conditions. In the submitted manuscript we suggest that Mig1 interaction with the DNA was a possible mechanism for Mig1 accumulation in the nucleus, though note we do not actually assert that the only possible cause, alongside interactions with proteins in the nucleus or cytoplasm. The reviewer is correct that the Zn finger deletion Mig1 localisation does not support this possible mechanism so the text has been amended to remove this.

The microfluidics experiments exploring Mig1-mCherry and PP7-2xGFP localisation were repeated using the Zn finger deletion strain and the results added to Figure 5 and referred to in the text. No accumulation of PP7 foci is seen in this strain as no change in repression happens upon changing extracellular glucose as the Mig1-mCherry ΔZnf cannot repress the *GYS1* gene^3^. This is similar to observations of Lin et al.^3^ They did see a small increase in transcriptional activity although with very high cell to cell variation using the same Mig1-mCherry ΔZnf strain but only when correlated with a burst of Msn2.

In Figure 3, the authors present data on the constrained diffusion of the foci due to nuclear anchoring of Mig1. If I understood properly, the deviation form the dashed line represents the constrained diffusion. However, I don't see a noticeable difference between the deviation observed for small or large foci or even cytoplasmic ones. Therefore I don't understand how this can be interpreted as a effect of the DNA anchoring of the Mig1 clusters. Again a ZF∆ Mig1 would be an interesting control to measure.

Molecular mobility within cells is, clearly, highly complex and the mode of diffusion dependent on the length and timescales over which it is observed. We analyse Mig1 cluster diffusion on very short timescales using the cumulative distribution function (CDF) analysis and over longer timescales up to ~50ms, and it is the latter to which the reviewer is referring. Over this timescale we find that Mig1 cluster diffusion is best accounted for by an anomalous diffusion model. We would hesitate to use the word “constrained” which might imply confined diffusion but we have removed the word “restrained” to avoid confusion. Anomalous diffusion is characterised by increasing deviation below the linear relation of means square displacement (MSD) with respect to time interval (tau), Brownian diffusion on a linear axes plot (see Figure 1 (responses)). The MSD vs tau plots in Figure 3 of the submitted manuscript are on log-log axes to better show the differences between foci; this has now been made clearer by rewording of the Figure 3 legend. The MSD vs tau plots are shown again in Author response image 1 and with linear axes in Author response image 1. The dotted line fits are all for anomalous diffusion, deviations from the fits do not represent constrained diffusion. The glucose (+) cytoplasmic and glucose (-) cytoplasmic and nuclear foci all lie on a similar line but the glucose (+) nuclear small and large foci are significantly lower, which we interpret as putative interaction with the DNA. This is further supported by CDF analysis which we also used with the Zn finger deletion strain, as suggested by the reviewer, showing only a single diffusing population – subsection “Cytoplasmic Mig1 is mobile but nuclear Mig1 has mobile and immobile states” has been amended to clarify this.

The authors present data on the in vitro oligomerisation of Mig1-GFP triggered by PEG addition. As a control, they use mGFP alone. First of all, this control should be performed with the same fluorescent protein for the TF bound and the GFP alone experiment. In addition, the number of measured fluorescent dots has to be the same in both cases. However 1000 dots where measured for the Mig1-GFP versus 100 for the GFP. Since the high stoichiometry clusters are found with <1% probability, the chance of observing a single high stoichiometry cluster is lower than 1 for the control experiment.

The same fluorescent protein for the TF bound and GFP alone has been used. We performed live cell Slimfield microscopy on both Mig1-mGFP and Mig1-eGFP. We observe clusters with both fluorophores and find no statistical difference between the measured stoichiometry (Figure 1—figure supplement 2), as stated in Results paragraph two. Despite no observable difference in vivobetween eGFP and mGFP, the in vitro clustering experiments were performed on Mig1-mGFP and mGFP to avoid any possibility of fluorescent protein mediated oligomerisation. Therefore we feel the correct fluorescent proteins have been used in each experiment. The reviewer is correct that more in vitroGFP foci should be included in the control experiments so these have been repeated and included in Figure 4—figure supplement 2.

I'm not a specialist of electron microscopy, however I find a few elements in this experiment puzzling. The authors state that they measured 150 cells and only in 10 of them did they see a cluster of Mig1 or Msn2. How many untagged cells did they measure to make sure that this phenomena is not present in untagged strains? Why don't we see individual Mig1 or Msn2 molecules labels with the gold particles in these images. Also no data of Mig1 nuclear cluster has been obtained. These low statistics makes me wonder about the quality of this dataset.

We agree with the reviewer, the statistics are low and the interpretation of these specific electron microscopy data should be weighted appropriately. With this caveat, the results are, however, consistent with the quantitative fluorescence imaging observations of clustering, and so do at least offer some supporting information from a complementary structural method of experimental investigation. But, these EM data would not be robust evidence in isolation from the light microscopy results due to the low statistics. To confirm, we looked at approximately the same number of parental strain cells (i.e. not containing any GFP) as the Mig1-GFP and Msn2-GFP strains (i.e. ca. 150 in total) – we have reworded the section in the Materials and methods (subsection “Immuno-gold electron microscopy”) to clarify this and also for transparency we have added specific emphasis in the main text concerning the technical challenges with the EM experiments resulting in relatively low statistics and that these should thus not be seen as robust proof of clustering alone (subsections “Mig1 clusters are spherical, a few tens of nm wide” and “The activator Msn2 also forms functional clusters”). We have also reworded the Materials and methods section to clarify that all of the EM images in the non-control (i.e. GFP containing) datasets contain evidence for gold labelling with at least 1 gold foci per cell (i.e. we do actually see individual gold particles), whereas we see no evidence of non-specific labelling with the control dataset. The value of 10 out of ~150 cells (i.e. ca. 7% of the total) refers to the number of cells in the GFP strain datasets which contains 2 or more gold foci separated by <50nm.

The principle challenges with the EM experiments were two-fold, one that the best effective binding efficiency to the primary (anti-GFP) antibody we were able to optimise was still relativity low once all of the EM sample prep stages had been completed (our estimate for overall binding efficiency for the combined primary and secondary labelling steps is maximally ~20%, which we have now clarified in the Materials and methods section), and two that, as the reviewer points out, there is sparse labelling in the nucleus itself. In the Discussion section we now include an estimate based on the volume fraction of the total haploid genome that the mean effective mesh size for DNA in the yeast nucleus is only a few tens of nm – this is comparable to the combined Stokes radii of primary and secondary antibodies used here and thus antibody probe accessibility into the nucleus is likely to be an issue in regards to successful labelling with gold particles. We have now included explicit mention of these issues in full in the revised text.

References

1. Chung, S. H. & Kennedy, R. A. Forward-backward non-linear filtering technique for extracting small biological signals from noise. *J. Neurosci. Methods* 40, 71–86 (1991).

2. Bendrioua, L. *et al.* Yeast AMP-activated protein kinase monitors glucose concentration changes and absolute glucose levels. *J. Biol. Chem.* 289, 12863–75 (2014).

3. Lin, Y., Sohn, C. H., Dalal, C. K., Cai, L. & Elowitz, M. B. Combinatorial gene regulation by modulation of relative pulse timing. *Nature* 527, 54–58 (2015).

4. Wollman, A. J. M. & Leake, M. C. Millisecond single-molecule localization microscopy combined with convolution analysis and automated image segmentation to determine protein concentrations in complexly structured, functional cells, one cell at a time. *Faraday Discuss.* 184, 401–24 (2015).